# Novel, provable algorithms for efficient ensemble-based computational protein design and their application to the redesign of the c-Raf-RBD:KRas protein-protein interface

Anna U. Lowegard[1,2]☯, Marcel S. Frenkel[3]☯, Graham T. Holt[1,2], Jonathan D. Jou[2], Adegoke A. Ojewole[1,2], Bruce R. Donald[2,3]*

**1** Program in Computational Biology and Bioinformatics, Duke University Medical Center, Durham, North Carolina, United States of America, **2** Department of Computer Science, Duke University, Durham, North Carolina, United States of America, **3** Department of Biochemistry, Duke University Medical Center, Durham, North Carolina, United States of America

☯ These authors contributed equally to this work.
* brd+pcb19@cs.duke.edu

**Data Availability Statement:** All of the computational experiments and code used and

## Abstract

The $K^*$ algorithm provably approximates partition functions for a set of states (e.g., protein, ligand, and protein-ligand complex) to a user-specified accuracy $\varepsilon$. Often, reaching an $\varepsilon$-approximation for a particular set of partition functions takes a prohibitive amount of time and space. To alleviate some of this cost, we introduce two new algorithms into the OSPREY suite for protein design: FRIES, a Fast Removal of Inadequately Energied Sequences, and *EWAK**, an Energy Window Approximation to $K^*$. FRIES pre-processes the sequence space to limit a design to only the most stable, energetically favorable sequence possibilities. *EWAK** then takes this pruned sequence space as input and, using a user-specified energy window, calculates $K^*$ scores using the lowest energy conformations. We expect FRIES/*EWAK** to be most useful in cases where there are many unstable sequences in the design sequence space and when users are satisfied with enumerating the low-energy ensemble of conformations. In combination, these algorithms provably retain calculational accuracy while limiting the input sequence space and the conformations included in each partition function calculation to only the most energetically favorable, effectively reducing runtime while still enriching for desirable sequences. This combined approach led to significant speed-ups compared to the previous state-of-the-art multi-sequence algorithm, *BBK**, while maintaining its efficiency and accuracy, which we show across 40 different protein systems and a total of 2,826 protein design problems. Additionally, as a proof of concept, we used these new algorithms to redesign the protein-protein interface (PPI) of the c-Raf-RBD:KRas complex. The Ras-binding domain of the protein kinase c-Raf (c-Raf-RBD) is the tightest known binder of KRas, a protein implicated in difficult-to-treat cancers. FRIES/*EWAK** accurately retrospectively predicted the effect of 41 different sets of mutations in the PPI of the c-Raf-RBD:KRas complex. Notably, these mutations include mutations whose effect had previously been incorrectly predicted using other computational methods. Next, we used FRIES/*EWAK** for prospective design and discovered a novel point mutation that improves binding

discussed in this manuscript are available from the Harvard Dataverse repository (https://doi.org/10.7910/DVN/VHIRNM). For new empirical designs, we recommend using the latest version of OSPREY available for free at http://www.cs.duke.edu/donaldlab/osprey.php. All computer code for the OSPREY system is also available on GitHub at https://github.com/donaldlab/OSPREY3, and is open-source and free.

**Funding:** This work is primarily support by the following grants from the National Institutes of Health (NIH): R01-GM078031 and R01-GM118543 to BRD. Website: www.nih.gov. In addition, AUL was partially supported by a PhRMA Foundation Pre Doctoral Fellowship in Informatics. Website: www.phrmafoundation.org. The funders had no role in study design, data collection and analysis, decision to publish, or preparation of the manuscript.

**Competing interests:** I have read the journal's policy and the authors of this manuscript have the following competing interests: BRD, MSF, and JDJ are founders of Ten63 Therapeutics. All other authors have declared that no competing interests exist.

of c-Raf-RBD to KRas in its active, GTP-bound state ($KRas^{GTP}$). We combined this new mutation with two previously reported mutations (which were highly-ranked by OSPREY) to create a new variant of c-Raf-RBD, c-Raf-RBD(RKY). FRIES/*EWAK\** in OSPREY computationally predicted that this new variant binds even more tightly than the previous best-binding variant, c-Raf-RBD(RK). We measured the binding affinity of c-Raf-RBD(RKY) using a bio-layer interferometry (BLI) assay, and found that this new variant exhibits single-digit nano-molar affinity for $KRas^{GTP}$, confirming the computational predictions made with FRIES/*EWAK\**. This new variant binds roughly five times more tightly than the previous best known binder and roughly 36 times more tightly than the design starting point (wild-type c-Raf-RBD). This study steps through the advancement and development of computational protein design by presenting theory, new algorithms, accurate retrospective designs, new prospective designs, and biochemical validation.

## Author summary

Computational structure-based protein design is an innovative tool for redesigning proteins to introduce a particular or novel function. One such function is improving the binding of one protein to another, which can increase our understanding of important protein systems. Herein we introduce two novel, provable algorithms, FRIES and *EWAK\**, for more efficient computational structure-based protein design as well as their application to the redesign of the c-Raf-RBD:KRas protein-protein interface. These new algorithms speed-up computational structure-based protein design while maintaining accurate calculations, allowing for larger, previously infeasible protein designs. Additionally, using FRIES and *EWAK\** within the OSPREY suite, we designed the tightest known binder of KRas, a heavily studied cancer target that interacts with a number of different proteins. This previously undiscovered variant of a KRas-binding domain, c-Raf-RBD, has potential to serve as a tool to further probe the protein-protein interface of KRas with its effectors and its discovery alone emphasizes the potential for more successful applications of computational structure-based protein design.

## Introduction

Computational structure-based protein design (CSPD) is an innovative tool that enables the prediction of protein sequences with desired biochemical properties (such as improved binding affinity). OSPREY (Open Source Protein Redesign for You) [1] is an open-source, state-of-the-art software package used for CSPD and is available at http://www.cs.duke.edu/donaldlab/osprey.php for free. OSPREY's algorithms focus on *provably* returning the optimal sequences and conformations for a given input model. In contrast, as argued in [2–7], stochastic, non-deterministic approaches [8–10] provide no guarantees on the quality of conformations, or sequences, and make determining sources of error in predicted designs very difficult.

When using OSPREY, the input model generally consists of a protein structure, a flexibility model (e.g., choice of sidechain or backbone flexibility, allowed mutable residues, etc.), and an all-atom pairwise-decomposable energy function that is used to evaluate conformations. OSPREY models amino acid sidechains using frequently observed rotational isomers or "rotamers" [11]. Additionally, OSPREY can also model continuous sidechain flexibility [12–15] along

with discrete and continuous backbone flexibility [16–19], which allow for a more accurate approximation of protein behavior [13, 16, 20–23]. The output produced by CSPD generally consists of a set of candidate sequences and conformations. Many protein design methods have focused on computing a global minimum energy conformation (GMEC) [14, 18, 24–28]. However, a protein in solution exists not as a single, low-energy structure, but as a thermodynamic ensemble of conformations. Models that only consider the GMEC may incorrectly predict biophysical properties such as binding [12, 20–23, 29–31] because GMEC-based algorithms underestimate potentially significant entropic contributions. In contrast to GMEC-based approaches, the $K^*$ algorithm [12, 29, 30] in OSPREY provably approximates the Boltzmann-weighted partition function for a protein state, thereby modeling the thermodynamic ensemble. When designing for binding affinity, this enables the designer to calculate the $K^*$ score—a ratio of the Boltzmann-weighted partition functions for a protein-ligand complex that estimates the association constant, $K_a$ (further detailed in the Section entitled "Computational materials and methods"). $BBK^*$ [32] is an efficient, multi-sequence design algorithm that calls the $K^*$ algorithm as a subroutine. Previous algorithms [12, 27, 29, 30, 33–35] that design for binding affinity using ensembles are linear in the size of the sequence space $N$, where $N$ is exponential in the number of simultaneously mutable residue positions. $BBK^*$ is the first provable ensemble-based algorithm to run in time sublinear in $N$, making it possible not only to perform $K^*$ designs over large sequence spaces, but also to enumerate a gap-free list of sequences in order of decreasing $K^*$ score.

OSPREY has been used successfully on several empirical, prospective designs including designing enzymes [12, 16, 22, 29, 36], resistance mutations [2, 37, 38], protein-protein interaction inhibitors [30, 39], epitope-specific antibody probes [40], and broadly-neutralizing antibodies [41, 42]. These successes have been validated experimentally *in vitro* and *in vivo* and are now being tested in several clinical trials [43–45]. However, while OSPREY has been successful in the past, as the size of protein design problems grows (e.g., when considering a large protein-protein interface), enumerating and minimizing the necessary number of conformations and sequences to satisfy the provable halting criteria in previous $K^*$-based algorithms [12, 29, 30] becomes prohibitive (despite recent algorithmic improvements [32]). The entire conformation space can be monumental in size and heavily populated with energetically unfavorable sequences and conformations. $EWAK^*$, an Energy Window Approximation to $K^*$, seeks to alleviate some of this difficulty by restricting the conformations included in each sequence's thermodynamic ensemble. $EWAK^*$ *guarantees* that each conformational ensemble contains *all* of the lowest energy conformations within an energy window of the GMEC for each design sequence. FRIES, a Fast Removal of Inadequately Energied Sequences, also mitigates this complexity problem by limiting the input sequence space to only the most favorable, low energy sequences. Previous algorithms have focused on optimizing for sequences whose conformations are similar in energy to that of the GMEC. In contrast, FRIES focuses on optimizing for sequences with energies better-than or comparable-to the wild-type sequence. FRIES *guarantees* that the restricted input sequence space includes all of the sequences within an energy window of the wild-type sequence, but excludes any potentially unstable sequences with significantly worse partition function values. Wild-type sequences are generally expected to be near-optimal for their corresponding folds [46]. Therefore, limiting the sequence space to sequences energetically similar to or better than the wild-type sequence is reasonable.

We compare $BBK^*$ with $K^*$ (henceforth referred to as $BBK^*$) to $BBK^*$ with $EWAK^*$ and FRIES (henceforth referred to as $EWAK^*$ and FRIES) to test our new methods. The implementation details of these algorithms involve some technical distinctions, which are discussed in S4 Text. Compared to the previous state-of-the-art algorithm $BBK^*$, FRIES and $EWAK^*$ improve runtimes by up to 2 orders of magnitude, FRIES decreases the size of the sequence space by up

to 2 orders of magnitude, and *EWAK*\* decreases the number of conformations included in partition function calculations by up to almost 2 orders of magnitude. These improvements are shown across 2,826 protein design problems spanning 40 different protein systems (see the Section entitled "Computational experiments" for more details).

As a proof of concept to further test these algorithms and our design approach, we used FRIES and *EWAK*\* to study the protein-protein interface (PPI) of KRas$^{GTP}$ in complex with its tightest-binding effector, c-Raf. As described in the Section entitled "Computational redesign of the c-Raf-RBD:KRas protein-protein interface," KRas is an important cancer target that has been heavily studied and exhibits a thoroughly optimized protein-protein interface in its interactions with its effectors [47–59]. For this study, we focused on the redesign of the c-Raf Ras-binding domain (c-Raf-RBD), the tightest known naturally-occurring binding partner of KRas, in complex with KRas$^{GTP}$ (c-Raf-RBD:KRas$^{GTP}$). First, our new algorithms successfully retrospectively predicted the effect on binding of mutations in the c-Raf-RBD:KRas$^{GTP}$ PPI even where other computational methods previously failed [60]. Next, we used FRIES/*EWAK*\* prospectively to predict the effect of novel, previously unreported mutations in the PPI of the c-Raf-RBD:KRas$^{GTP}$ complex. We then screened the top OSPREY-predicted c-Raf-RBD variants using a bio-layer interferometry (BLI) assay single-concentration screen. Looking at the dissociation rates, this screen suggested that one of our new computationally-predicted c-Raf-RBD variants—c-Raf-RBD(Y), a c-Raf-RBD that includes the mutation V88Y—exhibits improved binding to KRas$^{GTP}$. Next, we created a c-Raf-RBD variant, c-Raf-RBD(RKY), that included this new mutation, V88Y, together with two previously reported mutations [60], N71R and A85K. FRIES/*EWAK*\* computationally predicted that c-Raf-RBD(RKY) would bind more tightly to KRas$^{GTP}$ than any other variant. The single-concentration screen using BLI also suggested that c-Raf-RBD(RKY) binds more tightly to KRas$^{GTP}$ than the previously reported best variant [60]. The $K_d$ values for the most promising variants were measured using a BLI assay with titration which confirmed our computational predictions and that, to the best of our knowledge, the novel construct c-Raf-RBD(RKY) is the highest affinity variant ever designed, with single-digit nanomolar affinity for KRas$^{GTP}$ and binding roughly 36 times more tightly than the design starting point (wild-type c-Raf-RBD).

## Computational materials and methods

The $K^*$ algorithm's [12, 29, 30] $K^*$ score serves as an estimate of the binding constant, $K_a$, and is calculated by first approximating the Boltzmann-weighted partition function of each state: unbound protein (*P*), unbound ligand (*L*), and the bound protein-ligand complex (*C*). Each Boltzmann-weighted partition function $Z_x(\mathbf{s})$, $x \in \{P, L, C\}$, is defined as:

$$Z_x(\mathbf{s}) = \sum_{d \in \mathbf{Q}(\mathbf{s})} \exp(-E_x(d)/RT). \tag{1}$$

If **s** is any—generally amino acid—sequence of *n* residues, then $\mathbf{Q}(\mathbf{s})$ is the set of conformations defined by **s**, $E_x(d)$ is the minimized energy of a conformation *d* in state *x*, and *R* and *T* are the gas constant and temperature, respectively. Many protein design algorithms approximate these partition functions for each state using either stochastic [61–64] or provable [2, 12, 29–31, 33, 64] methods.

OSPREY's $K^*$ algorithm approximates these partition functions to within a user-specified $\varepsilon$ of the full partition function as defined in Eq (1) where *C*, *P*, and *L* refer to the protein-ligand complex, the unbound protein, and the unbound ligand, respectively. The binding affinity for

sequence **s** is defined as:

$$K_a(\mathbf{s}) = \frac{Z_C(\mathbf{s})}{Z_P(\mathbf{s})Z_L(\mathbf{s})}.$$
(2)

The $K^*$ algorithm provably approximates this binding affinity. This is enabled by the use of $A^*$ [4, 12, 26, 65], which allows for the gap-free enumeration of conformations in order of increasing lower bounds on energy [26]. However, enumerating a sufficient number of these conformations to obtain a guaranteed $\varepsilon$-approximation can be very time consuming because the set of all conformations $\mathbf{Q}(\mathbf{s})$ grows exponentially with the number of residues $n$. Also, the $K^*$ algorithm was originally [12, 29, 30] limited to computing a $K^*$ score for every sequence in the sequence space as defined by the input model for a particular design. However, $BBK^*$ [32] builds on $K^*$ and returns the top $m$ sequences along with their $\varepsilon$-approximate $K^*$ scores and runs in time sublinear in the number of sequences. That is, $BBK^*$ does not require calculating $\varepsilon$-approximate $K^*$ scores for (or even examining) every sequence in the sequence space before it returns the top sequences. Nevertheless, $BBK^*$ may spend unnecessary time and resources evaluating unfavorable sequences before deciding to prune them. These previous methods, while efficient, suffer from two practical drawbacks. First, some returned sequences exhibit a large $K^*$ score (i.e. are predicted to improve binding) due to a decrease in stability of the unbound states. These sequences are rarely desirable in practice, since decreasing protein stability can result in poor folding and aggregation. Second, the approximation error for some sequences is slow to approach epsilon which can lead to prohibitively slow designs.

To overcome the above limitations of $BBK^*$ and $K^*$, we introduce FRIES, a Fast Removal of Inadequately Energied Sequences, and $EWAK^*$, an Energy Window Approximation to $K^*$. These two algorithms limit the input sequence space and the number of conformations included in each partition function estimate when approximating a sequence's $K^*$ score to only the most energetically favorable options (see Fig 1). The FRIES/$EWAK^*$ approach limits the number of conformations that must be enumerated (see the Section entitled "FRIES limits the number of minimized conformations when approximating partition functions while maintaining accurate K* scores"), which leads to significant speed-ups (see the Section entitled "FRIES/$EWAK^*$ is up to 2 orders of magnitude faster than $BBK^*$") because each enumerated conformation must undergo an energy minimization step. This minimization step is relatively expensive, therefore, anything that reduces the number of minimized conformations while not sacrificing provable accuracy is desirable. For the importance of this minimization step to biological accuracy, see the discussions of continuous flexibility and its comparison to discrete flexibility in [4, 5, 7, 13, 14, 19]. $EWAK^*$ also maintains the advances made by $BBK^*$ including running in time sublinear in the number of sequences $N$ and returning sequences in order of decreasing $K^*$ score. FRIES and $EWAK^*$ are described in further detail in the Section entitled "Algorithms" below.

## Algorithms

**Fast removal of inadequately energied sequences (FRIES).**   Generally in protein design when optimizing a protein-protein interface (PPI) for affinity, the designer aims to improve the $K^*$ score of a variant sequence relative to the wild-type sequence, and, when performing a design targeting a similar fold, to minimally perturb the native structure. To accomplish this, FRIES guarantees to only keep sequences whose partition function values are not markedly worse than the wild-type sequence's partition function values for all of the design states (e.g. protein, ligand, and complex). How many orders of magnitude worse a particular sequence's partition function values are allowed to be is determined by a user-specified value $m$. The FRIES

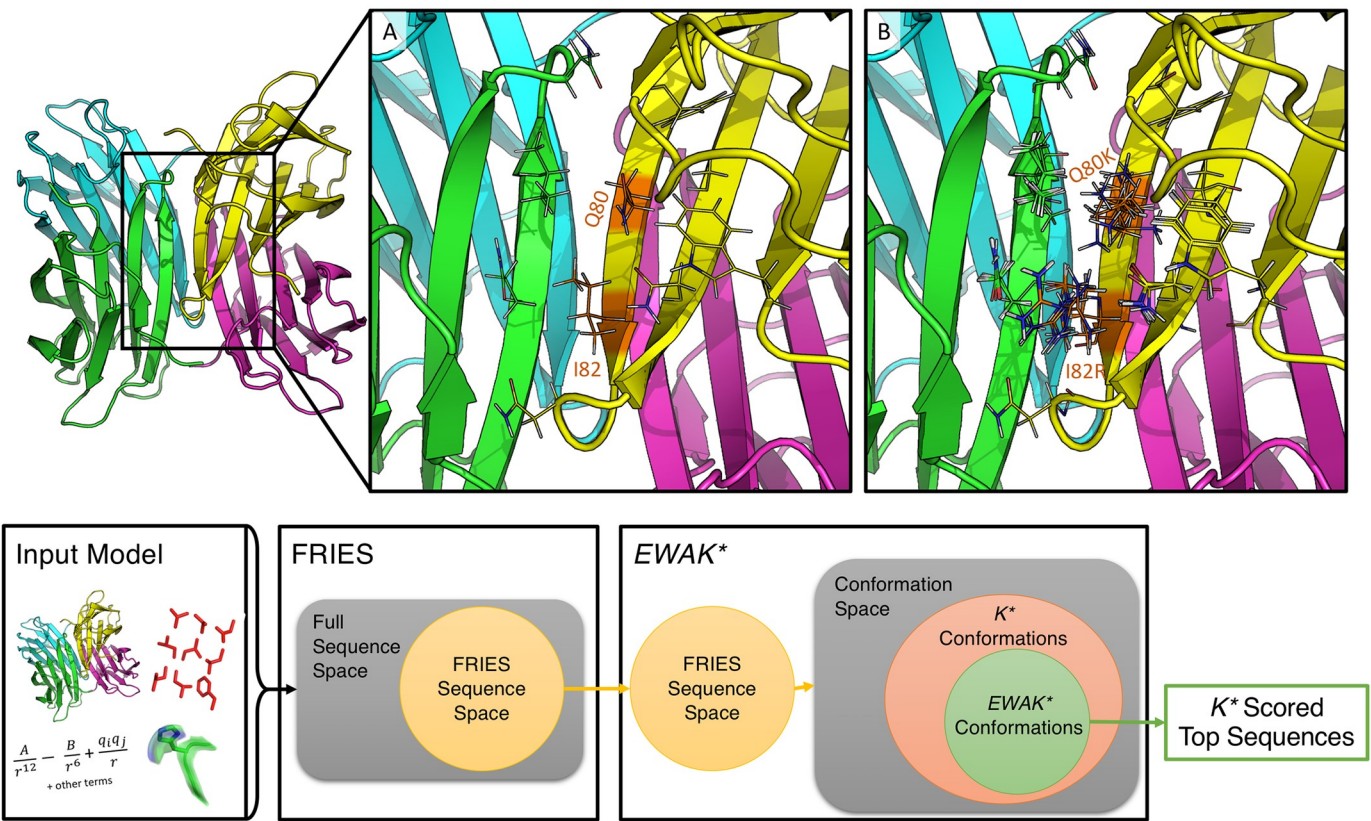

**Fig 1. Design example using the structure of the LecB lectin *Pseudomonas aeruginosa* strain PA14 (PDB ID: 5A6Y [67]) and the OSPREY workflow for FRIES/*EWAK**.** In the top panel, the full, 4 domain structure of lectin is shown on the left-hand side. (A) Zooming in on the region where domains A (green) and D (yellow) interact, showing the two mutable residues (Q80 and I82) along with the surrounding flexible shell of residues as lines. There were 11 flexible residues included in this design with Q80 and I82 allowed to mutate to all other amino acids except for proline. This design consisted of $8.102 \times 10^{11}$ conformations and 441 sequences. FRIES limited this space to $5.704 \times 10^{11}$ conformations and 206 sequences. FRIES/*EWAK** in combination reduced the amount of time taken by about 75% compared to *BBK**. FRIES alone was responsible for roughly 50% of this speed-up. (B) 10 low-energy conformations included in the thermodynamic ensemble of the design sequence with mutations Q80I and I82F. For this particular sequence, *BBK** minimized 10,664 conformations while *EWAK** minimized only 4,104 conformations. The bottom panel shows the general workflow for FRIES/*EWAK**. The workflow begins with the input model (as described in the Section entitled "Computational materials and methods"), which defines the design space for the first algorithm, FRIES. FRIES proceeds to prune the sequence space as described in the Section entitled "Fast Removal of Inadequately Energied Sequences (FRIES)" and as illustrated in the Venn diagram with the unpruned space shown as a yellow disk. Next, the remaining FRIES sequence space defines the conformation space (which contains multiple sequences as well as conformations) searched with *EWAK**. *EWAK** limits the conformations included in each partition function as described in the Section entitled "Energy Window Approximation to K* (*EWAK**)." *EWAK** generally searches over only a subset of the conformations (green area) that previous K*-based algorithms like *BBK** [32] search (orange area). *EWAK** then returns the top sequences based on decreasing K* score.

algorithm prunes sequences that exhibit massive decreases in partition function values that signal an increased risk of disturbing the native structure of the states in a given system. However, sequences with markedly worse, lower partition function values may be required when searching for, for example, resistance mutations, where positive and negative design are necessary [2, 37, 38]. Importantly, FRIES does still allow for sequences that may have lower, worse partition function values by allowing the user to specify how many orders of magnitude lower a candidate sequence's partition function is allowed to be relative to the wild-type sequence's partition function.

The following algorithm is applied to each of the three states (protein, ligand, and protein-ligand complex) independently. The resulting, filtered sequence space is determined by taking the intersection of the output from the algorithm for the three states. To prune the input sequence space, FRIES exploits $A^*$ over a *multi-sequence tree* (as is described and used in COMETS

[66]), which enjoys a fast sequence enumeration in order of lower bound on minimized energy. Each sequence $v$ in this *multi-sequence tree* [66] has a corresponding *single-sequence conformation tree*, viz., a tree that can be searched for the lowest energy conformations for a sequence $v$. FRIES first enumerates sequences (in order of energy lower bounds) in the *multi-sequence tree* until the wild-type sequence is found. Then, FRIES searches the wild-type's corresponding *single-sequence conformation tree* using $A^*$. The first conformation enumerated according to monotonic lower bound on pairwise minimized energy is then subjected to a full-atom minimization [30] to calculate the minimized energy of one of the wild-type sequence's conformations $E_{WT}$. It is worth noting that FRIES only descends into and searches the single-sequence conformation tree for the wild-type sequence in order to calculate the provable halting criteria for Eq 3. FRIES then continues enumerating sequences in the *multi-sequence tree* in order of increasing lower bound on minimized energy (as described in more detail in [66]) until the lower bound on the optimal conformational energy for a sequence $v$, $E_v^\ominus$, is greater than $E_{WT} + w$ where $E_{WT}$ is as described above and $w$ is a user-specified energy window value (see Fig 2). Any variant sequence $v$ with a lower bound on minimized energy $E_v^\ominus$ not satisfying the following criterion is pruned:

$$E_v^\ominus \leq E_{WT} + w. \tag{3}$$

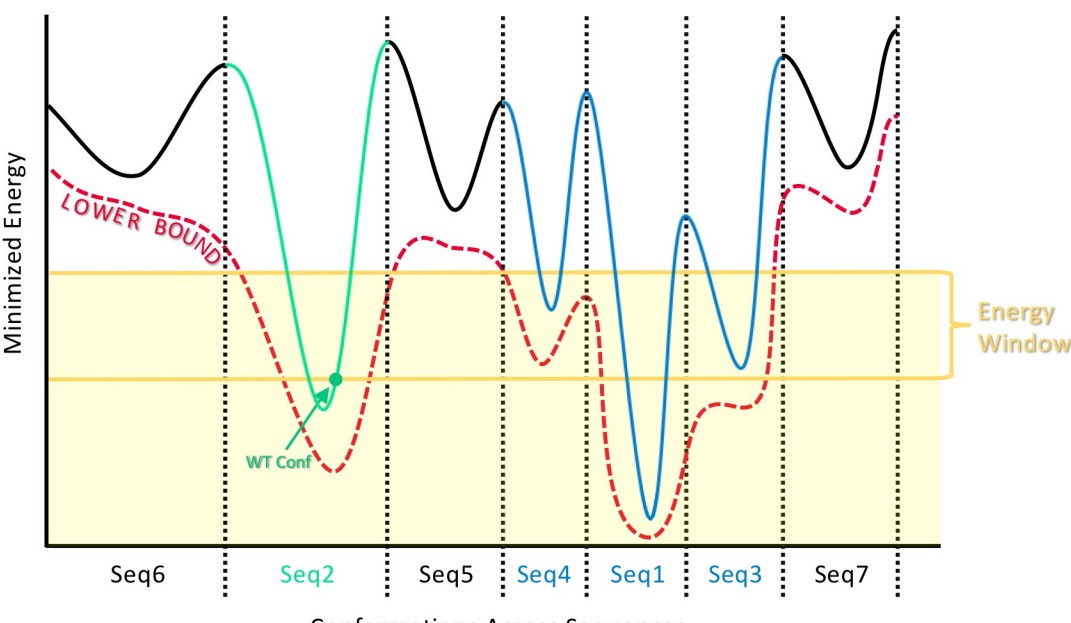

**Fig 2. How FRIES chooses which sequences to keep and which sequences to prune.** The solid curve represents the energy landscape of the conformation space that spans across, in this example, 7 different sequences (separated by dotted lines). Each sequence is labeled on the *x*-axis with an index indicating the order with which it is (or would be) enumerated with FRIES in order of increasing lower bound on minimized energy (red dotted curve). FRIES continues to enumerate in this way until it encounters the wild-type sequence (green), at which point FRIES calculates the minimized energy $E_{WT}$ of the conformation with the lowest lower bound on minimized energy for the wild-type sequence (marked with a green dot). $E_{WT}$ then becomes the baseline from which FRIES can provably enumerate all remaining sequences within some user-specified energy window $w$ (yellow lines). Finally, FRIES prunes the sequences with energies provably higher than $E_{WT} + w$ (black) and keeps the sequences that occur within the shaded yellow region (colored in blue and green). More sequences are also pruned according to their partition function values as described in the Section entitled "Fast Removal of Inadequately Energied Sequences (FRIES)" and Eq (4).

This criterion guarantees that the remaining, unpruned sequence space includes all sequences within an energy window of the wild-type sequence's energy. FRIES enumerates sequences in order of increasing lower bound on minimized energy. Therefore, it calculates an upper bound $q_v^{\oplus}$ on the partition function for each sequence $v$ by Boltzmann-weighting the lower bound on its energy $E_v^{\ominus}$ and multiplying it by the size of the conformation space for that particular sequence $|\mathbf{Q}(v)|$:

$$q_v^{\oplus} = |\mathbf{Q}(v)| \exp(-E_v^{\ominus}/RT). \tag{4}$$

The lower bound for the wild-type sequence $q_{WT}^{\ominus}$ is calculated by Boltzmann-weighting the minimized energy of the single conformation found during the sequence search for the wild-type sequence $E_{WT}$:

$$q_{WT}^{\ominus} = \exp(-E_{WT}/RT). \tag{5}$$

$q_{WT}^{\ominus}$ is a lower bound because, in the worst case, at least this one conformation will contribute to the partition function for the wild-type sequence. FRIES then uses these bounds to remove all of the sequences whose partition function value is not within some user-specified $m$ orders of magnitude of the lower bound on the wild-type partition function $q_{WT}^{\ominus}$. If the following criterion is not met, the sequence $v$ is pruned from the space:

$$\ln q_v^{\oplus} \leq \ln q_{WT}^{\ominus} + m. \tag{6}$$

FRIES prunes sequences for the protein, the ligand, and the protein-ligand complex independently, limiting the input sequence space to exclude unfavorable sequences for all of the states. The resulting smaller sequence space is subsequently used as input for $EWAK^*$. The set of sequences remaining is guaranteed to include all of the sequences within a user-specified energy window $w$ of the wild-type sequence that also satisfy the partition function criterion given in Eq (4). Importantly, FRIES can be used to limit the size of the input sequence space in this fashion for any of the protein design algorithms available within OSPREY.

**Energy window approximation to K\* (EWAK\*).**   After reducing the size of the input sequence space using FRIES, as described in the Section entitled "Fast Removal of Inadequately Energied Sequences (FRIES)," $EWAK^*$ proceeds by using a variation on an existing algorithm: $BBK^*$ (described in [32]). The crucial difference between $BBK^*$ and $EWAK^*$ is that with $EWAK^*$ the ensemble of conformations used to approximate each $K^*$ score is limited to those within a user-specified energy window of the GMEC for each sequence. This guarantees to populate the partition function for a particular sequence and state with all of the lowest, most-favorable conformations (that fall within the user-specified energy window). Limiting the partition functions to only these energetically favorable conformations can effectively reduce runtime while still enriching for desirable sequences. These conformations often account for the majority of the full $\varepsilon$-approximate partition function (see the Section entitled "Computational materials and methods") in traditional $K^*$ calculations [12]. Hence, $EWAK^*$ also empirically enjoys negligible loss in accuracy of $K^*$ scores (see the Sections labeled "$EWAK^*$ limits the number of minimized conformations when approximating partition functions while maintaining accurate K\* scores" and "FRIES/$EWAK^*$ retrospectively predicted the effect mutations in c-Raf-RBD have on binding to KRas"). $EWAK^*$ retains the beneficial aspects of $BBK^*$, including returning sequences in order of decreasing predicted binding affinity and running in time sublinear in the number of sequences. For a discussion of the relationship between $\varepsilon$ and the energy window $w$, the interested reader is invited to refer to the SI.

## Computational experiments

We implemented FRIES/*EWAK** in the OSPREY suite of open source protein design algorithms [1]. FRIES was tested on 2,662 designs that range from an input sequence space size of 441 to 10,164 total sequences. The size of the reduced input sequence space produced by FRIES was compared to the size of the full input sequence space size for each design. For these tests, FRIES returned every sequence within 8 kcal/mol of the wild-type sequence and was set to include only those sequences that are at most 2 orders of magnitude worse in partition function value than the wild-type. The results for these tests are described in the Section entitled "FRIES can reduce the size of the input sequence space by more than 2 orders of magnitude while retaining the most favorable sequences." Computational experiments were also run comparing FRIES/*EWAK** with the previous state-of-the-art algorithm in OSPREY: *BBK** [32]. Using *BBK** and FRIES/*EWAK**, we computed the top 5 best binding sequences for 167 different designs to compare the running time of *BBK** vs. FRIES/*EWAK**. FRIES was limited to sequences within 4 kcal/mol of the wild-type sequence that are at most 2 orders of magnitude worse in partition function values than the wild-type. The *EWAK** partition function approximations were limited to conformations within an energy window of 1 kcal/mol of the GMEC for each sequence. *BBK** was set to return the top 5 sequences with an accuracy of $\varepsilon = 0.68$ (as was described in [32]). Using these same *EWAK** and *BBK** parameters, we also compared the change in the size of the conformation space necessary to compute an accurate $K^*$ score for *BBK** vs. *EWAK** for 661 partition functions from 161 design examples. The results for these tests are described in the Sections labeled "FRIES/*EWAK** is up to 2 orders of magnitude faster than *BBK**" and "FRIES can reduce the size of the input sequence space by more than 2 orders of magnitude while retaining the most favorable sequences". The number of conformations that undergo minimization (as described in [12–15]) for each partition function calculation with *EWAK** was also compared across different energy window sizes for 350 partition function calculations from 87 design examples. These partition function calculations were compared to *BBK**'s partition function calculations with a demanded accuracy of $\varepsilon = 0.10$. This smaller $\varepsilon$ allowed for more accurate approximations of the $K^*$ scores. The results for these tests are described in the Section entitled "FRIES can reduce the size of the input sequence space by more than 2 orders of magnitude while retaining the most favorable sequences".

Every design included a set of mutable residues along with a set of surrounding flexible residues (see Fig 1 for an example). All of these residues were allowed to be continuously flexible [12–15]. The designs were selected from 40 different protein structures (listed in S1 Table and also used in [32, 68]), and were run on 40-48 core Intel Xeon nodes with up to 200 GB of memory.

## Computational results

### FRIES can reduce the size of the input sequence space by more than 2 orders of magnitude while retaining the most favorable sequences

The number of remaining sequences after FRIES was compared to the size of the complete input sequence space. In the best case, when using FRIES, the sequence space was decreased by more than 2 orders of magnitude and the conformation space was decreased by just over 4 orders of magnitude. The sequence space was reduced an average of 49% and the conformation space was reduced an average of 40%. These results are broken down further in Fig 3.

### FRIES/*EWAK** is up to 2 orders of magnitude faster than BBK*

The overall runtime was compared between *BBK** and FRIES/*EWAK**. FRIES/*EWAK** was an average of 62% faster than *BBK** on 167 example design problems. FRIES removed unfavorable

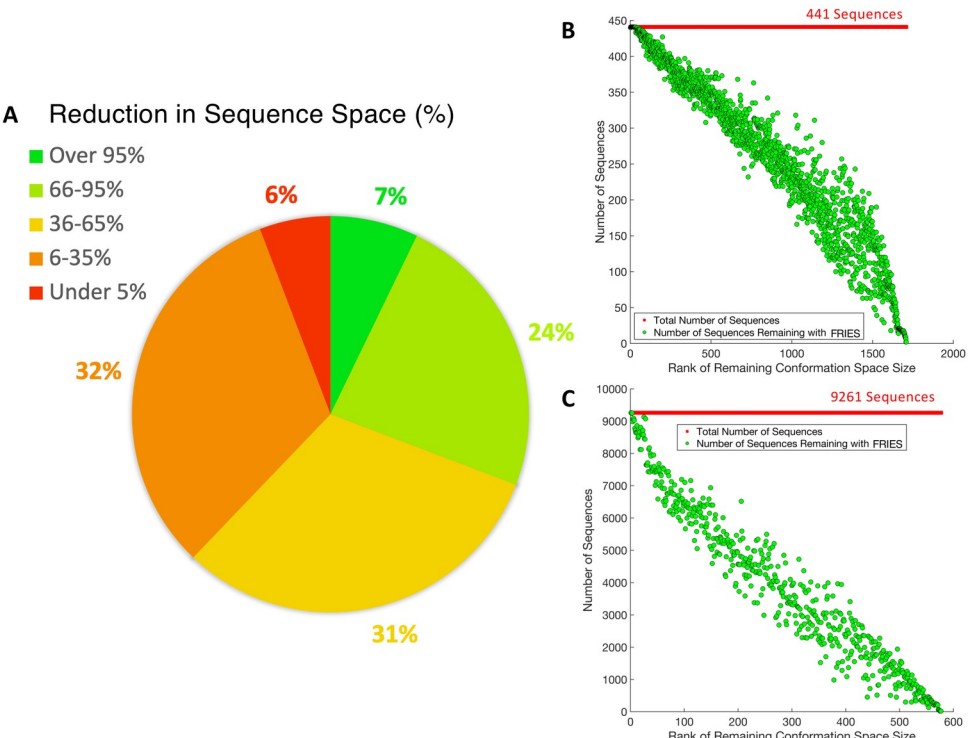

**Fig 3. Reduction in input sequence space size using FRIES.** (A) A pie chart representing the reduction in the sequence space in percentages across all 2,662 designs. 7% of the designs had a reduction in sequence space over 95%, 24% of the designs had a reduction in sequence space between 66-95%, 31% of the designs had a reduction in sequence space between 36-65%, 32% of the designs had a reduction in sequence space between 6-35%, and 6% of the designs had a reduction in sequence space under 5%. (B) and (C) plot the number of sequences remaining after using FRIES starting with 441 and 9,261 sequences total, respectively. The number of sequences remaining for each design are sorted in order of decreasing size of the remaining conformation space after FRIES.

sequences (as described in the Section entitled "Fast Removal of Inadequately Energied Sequences (FRIES)") from the search space for 156 out of the 167 design problems. For the cases described in the Section entitled "Computational experiments," FRIES/*EWAK** performed consistently faster than *BBK** (in 92% of the design examples) as shown in Fig 4, Panel A. The longest running *BBK** design problem took nearly 8 days, whereas FRIES/*EWAK** completed the same example in just under 2 hours. In contrast, the design problem that took the longest for FRIES/*EWAK** out of the 167 tested only required about 22 hours (the same design took *BBK** just over 178 hours).

## *EWAK** limits the number of minimized conformations when approximating partition functions while maintaining accurate K* scores

We examined 661 $K^*$ score calculations, and concluded that the total number of conformations minimized to approximate the $K^*$ score was decreased by an average of 27%. In the best case the number of conformations minimized to approximate the $K^*$ score was decreased by 93%. These results are plotted in Fig 4, Panel B. Even though the partition function approximations were limited to a smaller conformation space with *EWAK**, the $K^*$ scores did not differ by more than 0.2 orders of magnitude between *EWAK** and *BBK** for these 661 example $K^*$ score calculations.

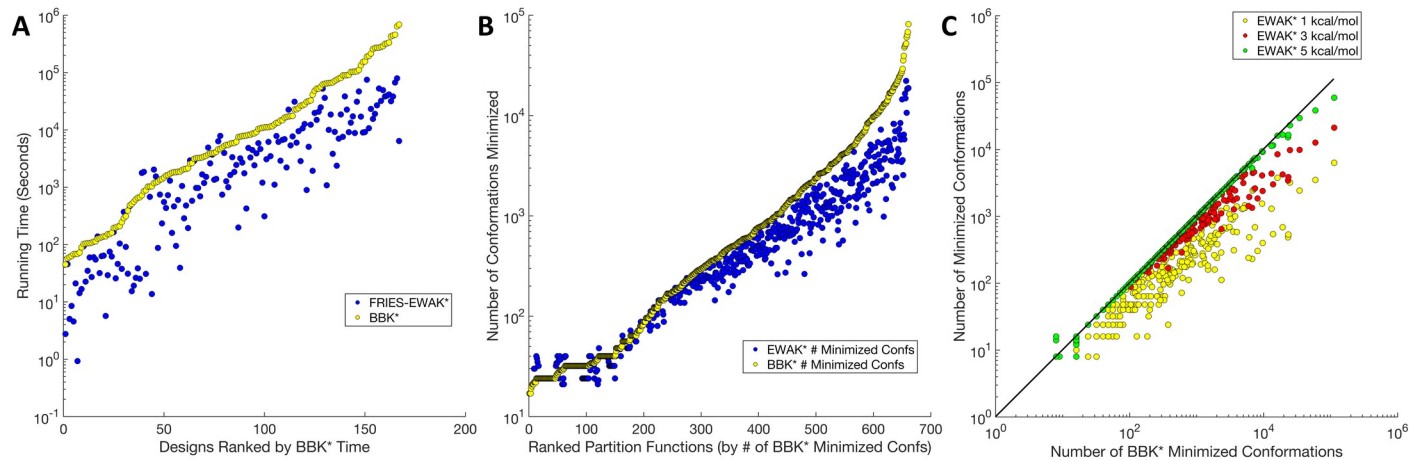

**Fig 4. Comparing runtimes and the number of minimized conformations between FRIES/*EWAK** and *BBK** for a variety of designs.** (A) A plot of the runtime in seconds (the *y*-axis is on a log scale) for FRIES/*EWAK** (blue dots) and *BBK** (yellow dots) for 167 design examples. Each point represents one design and is plotted in increasing order of *BBK** running time. FRIES/*EWAK** was faster than *BBK** 92% of the time with an average improvement of 62% over *BBK** and a maximum improvement of 2.2 orders of magnitude. This improvement was evident in (A) since the blue dots (FRIES/*EWAK** times) fall mostly below the yellow dots (*BBK** times). (B) A plot of the number of conformations minimized (*y*-axis is on a log scale) for 661 partition function calculations from 161 design examples. The number of conformations minimized by *EWAK** (blue dots) was less than the number of conformations minimized by *BBK** (yellow dots) in 68% of these cases, as is evidenced by the blue dots landing mostly below the yellow dots. In the best case, *EWAK** decreased the number of conformations by 1.1 orders of magnitude. The average percent reduction in the number of minimized conformations was 27%. (C) Each dot represents a calculated partition function. Yellow dots are partition functions limited to within a 1.0 kcal/mol window of the GMEC, red dots are partition functions limited to a 3.0 kcal/mol window of the GMEC, and green dots are partition functions limited to within a 5.0 kcal/mol window of the GMEC. These dots are plotted according to the number of minimized conformations required for each corresponding *BBK** partition function calculation. The solid black line represents the number of *BBK** minimized conformations, so dots that fall below the black line represent examples that required fewer minimized conformations than with *BBK**. As they approach the 5.0 kcal/mol window, the dots begin to converge with the *BBK** line. However, as the number of *BBK** minimized conformations rises beyond $\sim 10^4$, even the green dots drop below the *BBK** line.

A total of 350 of these 661 partition functions were subsequently re-estimated using *BBK** with a more accurate, stringent $\varepsilon$ value of 0.1 and using *EWAK** with varied energy windows: 1.0 kcal/mol, 3.0 kcal/mol, and 5.0 kcal/mol. We examined the number of conformations minimized for each complex partition function calculation across the examples. When using 1.0 kcal/mol, *EWAK** minimized up to 1.7 orders of magnitude fewer conformations (see Fig 4, Panel C for more details). Despite this decrease in the number of included conformations, *EWAK** reported accurate *K** scores. The largest difference in scores between *BBK** and *EWAK** was 0.3 orders of magnitude. This indicates that *EWAK** retains accuracy when compared to previous provable algorithms, which have been extensively validated using experimental measurements of binding, crystal structures, and NMR structures on a variety of systems [22, 30, 36–38, 40–42]. The accuracy of *EWAK** is explored further in the Section entitled "FRIES/*EWAK** retrospectively predicted the effect mutations in c-Raf-RBD have on binding to KRas," where we perform additional retrospective validation against experimental measurements.

## Computational redesign of the c-Raf-RBD:KRas protein-protein interface

We previously showed, investigating 58 mutations across 4 protein systems, that OSPREY can accurately predict the effect of mutations on PPI binding [1]. Herein, we tested the biological accuracy of the new modules FRIES and *EWAK** after adding them to OSPREY in the case of a particular system: c-Raf-RBD in complex with KRas. The c-Raf Ras-binding domain (c-Raf-RBD) is a small self-folding domain that does not include the kinase signaling domains normally present in c-Raf. The c-Raf-RBD normally binds to KRas when KRas is GTP-bound

(KRas$^{GTP}$). KRas has been implicated in difficult-to-treat cancers such as pancreatic ductal adenocarcinoma (PDAC) and has therefore been thoroughly studied [47, 47, 48, 48, 49, 49–55, 55, 56, 56–60, 69, 70]. So, to further verify the accuracy and utility of FRIES/*EWAK**, we focused on this already heavily optimized PPI between KRas$^{GTP}$ and one of its many effectors, c-Raf-RBD. First, in the Section entitled "FRIES/*EWAK** retrospectively predicted the effect mutations in c-Raf-RBD have on binding to KRas," we retrospectively investigated previously reported mutations in the c-Raf-RBD [48, 49, 60] and how they effect the binding of c-Raf-RBD to KRas. This retrospective study lays the groundwork for the prospective study we present that investigates novel mutations. So, following the retrospective study, we computationally redesigned the PPI using FRIES/*EWAK** in search of new c-Raf-RBD variants with improved affinity for KRas$^{GTP}$ (see the Section entitled "Prospective redesign of the c-Raf-RBD:KRas protein-protein interface toward improved binding" for details). To perform these computational designs, we first made a homology model of c-Raf-RBD bound to KRas$^{GTP}$ (see S1 Text for details).

## FRIES/*EWAK** retrospectively predicted the effect mutations in c-Raf-RBD have on binding to KRas

Each previously reported c-Raf-RBD variant [48, 49, 60] was tested computationally using FRIES/*EWAK** by calculating a $K^*$ score, a computational approximation of $K_a$, for each variant along with its corresponding wild-type sequence. A percent change in binding was then calculated by comparing the variant's $K^*$ score to the corresponding wild-type sequence's $K^*$ score. The $\log_{10}$ of this value was then calculated and normalized to the wild-type by subtracting 2. A similar procedure was completed using the reported experimental data in order to easily compare the computationally predicted effect with the experimentally measured effect. The resulting value, called Δb, represents the change in binding. If a variant has a Δb less than 0, it is predicted to decrease binding. If a variant has a Δb greater than 0, it is predicted to increase binding. Δb values that are roughly equivalent to 0 indicate variants that have little to no effect on binding since the wild-type sequence was normalized to 0. The Δb values for the 41 computationally tested variants were plotted and compared to experimental values in Fig 5 (a table of these values is also presented in S2 Table).

Out of the 41 variants tested (see S2 Table), *EWAK** predicted the experimentally-reported effect (increased vs. decreased binding) correctly in 38 cases. The three designs where the effect was predicted incorrectly are marked with a star in Fig 5. To make these predictions, the corresponding computational designs ranged in size from single point mutations up to 6 simultaneous mutations. Results are outlined in Fig 5 and data is presented in S2 Table. The Pearson's *r* of the Δb values when comparing the experimental data to the computational predictions is 0.64. Furthermore, the Spearman's $\rho$ value—a measure of the correlation between two sets of rankings—when comparing the experimental data to the computational predictions is 0.81. This $\rho$ value indicates that not only can *EWAK** correctly predict the effect of a particular set of mutations, but that *EWAK** also does a good job ranking the variants in order according to change in binding upon mutation (see Fig 6). We emphasize Spearman's $\rho$ here as opposed to a Pearson's correlation since our current designs likely underestimate entropic contributions to binding due to solvent entropy, backbone entropy, and rotating methyl groups. Nevertheless, by explicitly modeling side-chain configurational entropy, our method considers more conformational entropy than GMEC-based methods—in [1, 38] large changes in $K^*$ score corresponded to significant changes in energy, and rankings correlated well with experimental binding measurements. The Spearman's $\rho$ for the study presented here is comparable to the

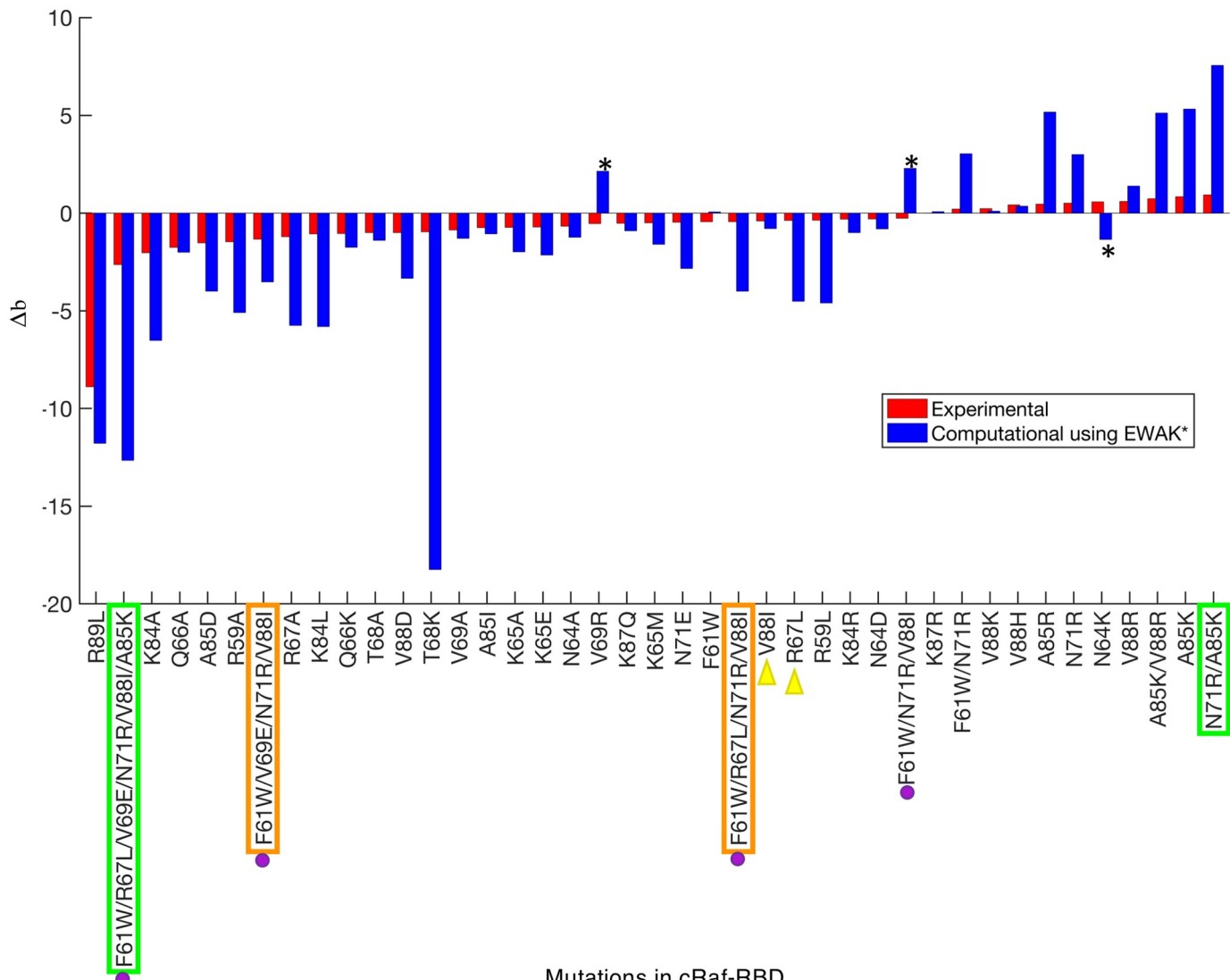

**Fig 5. Predicting the effect of mutations in c-Raf-RBD on binding with KRas.** Each bar represents either the experimental (red) or computationally predicted (blue) effect each variant has on binding. The bars are sorted in increasing order of Δb value (see the Section entitled "ғʀɪᴇs/*EWAK** retrospectively predicted the effect mutations in c-Raf-RBD have on binding to KRas") of the experimental (red) bars. If the Δb value is less than 0, binding decreases. If the Δb value is greater than 0, binding increases. If the Δb value is close to 0, the effect is neutral. Quantitative values of $K^*$ tend to overestimate the biological effects of mutations (leading to the much larger blue bars) due to the limited nature of the input model compared to a biologically accurate representation. However, $K^*$ in general does a good job ranking variants, as can be seen here in Fig 6, in [1], and in [38]. Out of the 41 variants listed on the *x*-axis, only 3 were predicted incorrectly (marked with black asterisks) by *EWAK**. In terms of accuracy, *BBK** performed very similarly to *EWAK**, however, in 2 cases (marked with green boxes), *BBK** ran out of memory and was unable to calculate a score. *BBK** also did not return values for the 2 variants marked with orange boxes. The variants marked with purple dots were tested in [60] experimentally—not computationally—and decreased binding of c-Raf-RBD to KRas^GTP was observed, which *EWAK** was able to predict correctly. The two variants marked with yellow triangles were computationally predicted in [60] to improve binding of c-Raf-RBD to KRas^GTP. However, the experimental validation in [60] showed that these variants exhibit decreased binding, which *EWAK** accurately predicted.

values for other PPI systems when using ᴏsᴘʀᴇʏ [1, 38]. Furthermore, an accurate ranking can guide an experimental lab in choosing the rank order in which to test computational predictions [2, 12, 16, 22, 29, 30, 36–42].

*BBK** produced similarly accurate results, but took up to 10 times longer and failed to produce results in 4 cases. In particular, in 2 cases (marked in green in Fig 5), *BBK** ran out of

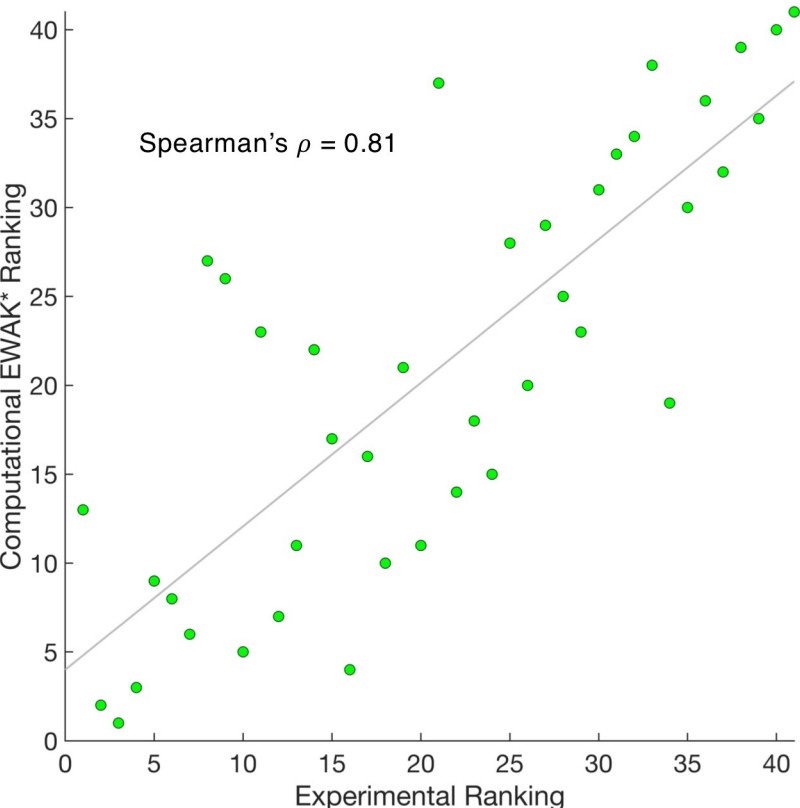

**Fig 6. Comparing the computational *EWAK*\* ranking with the experimental ranking for 41 c-Raf-RBD variants binding to KRas.** Each green dot represents a variant of c-Raf-RBD and is plotted according to the experimental ranking along with the corresponding computational ranking of its binding to KRas. A least squares fit line is shown in gray. Calculating the Pearson correlation coefficient between the two sets of rankings yields a Spearman's $\rho$ of 0.81.

memory. These cases in particular serve as examples of large designs where *EWAK*\* outperforms *BBK*\* and highlight the utility of FRIES/*EWAK*\* when considering larger designs. In the 2 other cases (marked in orange in Fig 5), *BBK*\* failed to return a result for the requested sequence in the top 5 reported sequences. This illustrated how *EWAK*\* and FRIES are particularly helpful when performing these types of bigger designs that contain more simultaneous mutations and more flexible residues.

Finally, we compared our predictions to the interesting biological predictions in [60]. It is unclear how many mutants were computationally evaluated, but the authors do report computational predictions for 6 point mutations. Of those, point mutants R67L, N71R, and V88I were predicted to improve the intermolecular interactions between c-Raf-RBD and KRas$^{\text{GTP}}$. However, experiments found that R67L and V88I actually reduced the binding of c-Raf-RBD to KRas$^{\text{GTP}}$ [48, 60]. In contrast to [60], *EWAK*\* accurately predicted that these mutations decrease binding of c-Raf-RBD to KRas$^{\text{GTP}}$. For a more detailed view of one of these designs, V88I, see Fig 7. Additionally, a number of mutations were combined and experimentally tested in [60]. Unfortunately, none of these variants improved binding to either KRas$^{\text{GTP}}$ or KRas$^{\text{GDP}}$, which FRIES/*EWAK*\* correctly predicted computationally (see Fig 5). In [60], the authors do not present any computational predictions for these combined variants, but our results show that a computational prediction using OSPREY's *EWAK*\* would have saved the time and resources taken to experimentally test these variants.

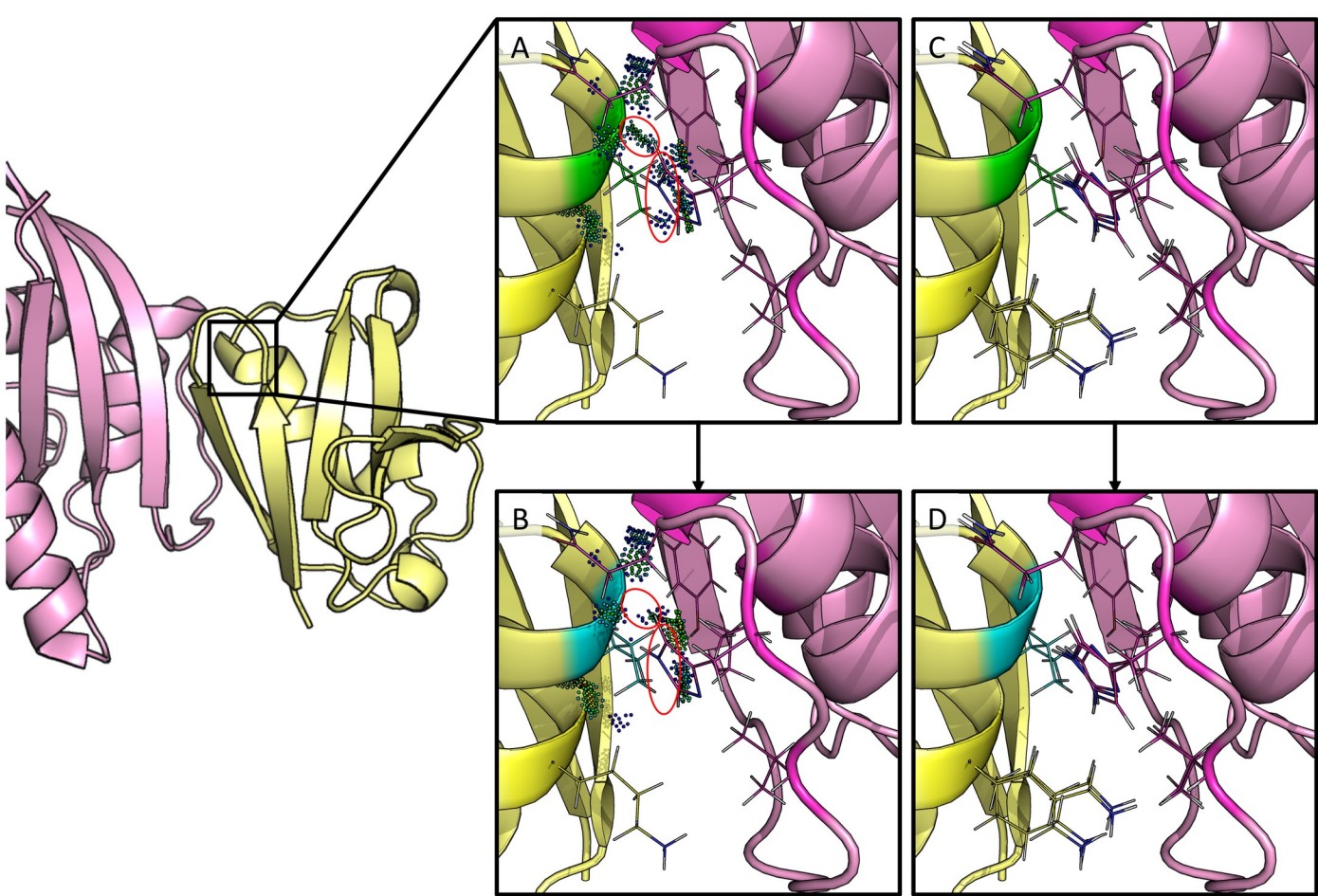

**Fig 7. Redesign of c-Raf-RBD residue position 88 from valine to isoleucine.** The left-hand side shows c-Raf-RBD (yellow) in complex with KRas (pink). Panels (A-D) zoom in on one particular design at residue position 88 and are rotated 180˚. Residue position 88 has a valine in the native, wild-type sequence (panels A & C) which was redesigned to an isoleucine (panels B & D). A mutation to isoleucine at this position was computationally predicted by *EWAK** to decrease the binding of c-Raf-RBD to KRas$^{GTP}$. This was experimentally validated in [60], where the authors incorrectly computationally predicted the effect of this particular mutation on the binding of c-Raf-RBD to KRas$^{GTP}$. (A) The wild-type residue (valine) is shown in green with dots that indicate molecular interactions [71] with the surrounding residues (residues allowed to be flexible in the design are shown as lines). (B) The mutant residue (isoleucine) is shown in blue with dots that indicate molecular interactions [71] with the surrounding residues (residues allowed to be flexible in the design are shown as lines). Contacts made by the wild-type valine residue (circled dots in (A)) were lost upon mutation to isoleucine (circled space in (B)). (C & D) A set of 10 low-energy conformations that were included in the corresponding partition function calculation are shown for the wild-type (green) and the variant (blue).

## Prospective redesign of the c-Raf-RBD:KRas protein-protein interface toward improved binding

The ability to accurately predict the effect mutations have on the binding of c-Raf-RBD to KRas$^{GTP}$ (see the Section entitled "FRIES/*EWAK** retrospectively predicted the effect mutations in c-Raf-RBD have on binding to KRas") gave us confidence in the *EWAK** algorithm's ability to predict new mutations in this interface toward a c-Raf-RBD variant that exhibits an even higher affinity for KRas$^{GTP}$ than previously reported variants which focused on targeting KRas$^{GDP}$ [60]. Therefore, to do a prospective study, we computationally redesigned 14 positions in c-Raf-RBD in the c-Raf-RBD:KRas PPI to identify promising mutations. After extending OSPREY to include FRIES and *EWAK**, 14 different designs were completed where each design included 1 mutable position that was allowed to mutate to all amino acid types except for proline. Each design also included a set of surrounding flexible residues within roughly 4 Å

**Table 1. Computational predictions by OSPREY/FRIES/*EWAK** that were selected for experimental validation.** Each row of the table shows the results of the redesign of a residue position in c-Raf-RBD in the c-Raf-RBD:KRas PPI that were also selected for experimental validation (all of the computational results are listed in S3 Table). The table contains the values for upper and lower bounds on log($K^*$) values (the calculation of these bounds is described in detail in [32]). Mutations highlighted in yellow, blue, and pink were selected for experimental testing and validation. The two residues highlighted in blue are the best previously discovered [60] mutations that improve binding (independently and additively) and are included in our tightest binding variant, c-Raf-RBD(RKY) (Figs 8, 9 and 10). The variants highlighted in yellow are, to the best of our knowledge, never-before-tested variants that are predicted to increase the binding of c-Raf-RBD to KRas$^{GTP}$. The variant highlighted in pink was selected for experimental testing to act as a mutation predicted to be comparable to wild-type to test how accurately OSPREY predicted the effects of these mutations.

| Mutation | Lower Bound log($K^*$) | Upper Bound log($K^*$) |
|----------|------------------------|------------------------|
| T57M | 3.43 | 3.46 |
| T57 | 3.82 | 3.92 |
| T57K | 5.01 | 5.07 |
| N71 | 7.25 | 7.49 |
| N71R | 9.66 | 10.10 |
| A85 | 26.3 | 26.9 |
| A85K | 30.7 | 32.3 |
| K87 | 13.4 | 14.1 |
| K87Y | 14.1 | 14.2 |
| V88 | 16.5 | 16.6 |
| V88Y | 17.3 | 17.6 |
| V88F | 18.0 | 18.2 |

of the mutable residue. These designs were run using FRIES and *EWAK** and included continuous flexibility [12–15]. FRIES was first used to limit each design to only the most favorable sequences (as described in the Section entitled "Fast Removal of Inadequately Energied Sequences (FRIES)") and then *EWAK** was used to estimate the $K^*$ scores (as described in the Section entitled "Energy Window Approximation to K* (*EWAK**)"). We report the upper and lower bounds on the *EWAK** score for each design in Table 1 and S3 Table, where the listed sequences are those that were not pruned during the FRIES step. From these results, the predicted binding effect (increased vs. decreased) was determined based on comparing each variant's $K^*$ score to its corresponding wild-type $K^*$ score. We then selected 5 novel point mutations—that to our knowledge are not reported in any existing literature—for experimental validation (see Table 1). It is worth noting that these 5 point mutations were selected out of an initial 294 possible mutations. We limited our experimental validation to only these 5 new mutations and 2 previously reported mutations. This greatly reduced the amount of resources necessary for experimental validation compared to testing all 294 possibilities. Of the mutations selected, T57M was selected to act as a variant that we computationally predicted to be comparable to wild-type. This variant was included to further verify the accuracy of OSPREY's predictions. On the other hand, some of OSPREY's top predictions were excluded, for instance, T57R (included in S3 Table) was not selected for experimental testing because it has an unsatisfied hydrogen bond as evidenced in the structures calculated by OSPREY. Another example is position V69 where 3 different mutations are predicted to improve binding, however, this position was included in our retrospective study (see the Section entitled "FRIES/*EWAK** retrospectively predicted the effect mutations in c-Raf-RBD have on binding to KRas" and Fig 5) and was 1 of only 3 positions where OSPREY incorrectly predicted the effect of the mutation. Therefore, we do not believe that the scores accurately represent the effect the mutations will have in these few cases. Other excluded top predictions (see S3 Table) displayed similar characteristics or have been reported and tested previously [48, 49, 60]. One special case that is not

shown in our experimental validation below is V88W which caused poor expression of c-Raf-RBD so we were unable to test it.

## Experimental validation of mutations in the c-Raf-RBD:KRas protein-protein interface

The mutations selected (highlighted in Table 1) from computational design were first screened using a bio-layer interferometry (BLI) single concentration assay (see the Section entitled "Bio-layer interferometry (BLI) dissociation rate and response screening" below). For this assay, we plotted response vs. dissociation rate constant (see Fig 9). This allowed us to quickly obtain a qualitative probe of c-Raf-RBD variant binding to KRas. It has been shown that off-rate measurements correlate to overall binding affinity [72–74]. A potential pitfall of depending only on off-rate observations is the potential for a slow off-rate to be paired with a slow on-rate, resulting in lower than expected affinity. Results from this initial single-concentration BLI screen (see Fig 8) suggested that, contrary to the computational predictions, the T57K and V88F variants decrease binding, whereas the T57M and K87Y mutations both have a roughly neutral effect on binding, which is consistent with the computational predictions. The final computationally predicted point mutant, V88Y, improves binding a comparable amount to the improvement seen with A85K or N71R, two previously reported variants that improve binding as correctly predicted by OSPREY and also experimentally tested herein. With the discovery of this new variant containing the point mutant V88Y (referred to herein as c-Raf-RBD(Y)) the next natural step was to combine it with the mutations found in the best reported variant, N71R and A85K (referred to herein as c-Raf-RBD(RK)). Therefore, we also included the double-mutant, c-Raf-RBD(RK), and the new triple-mutant—which contains N71R,

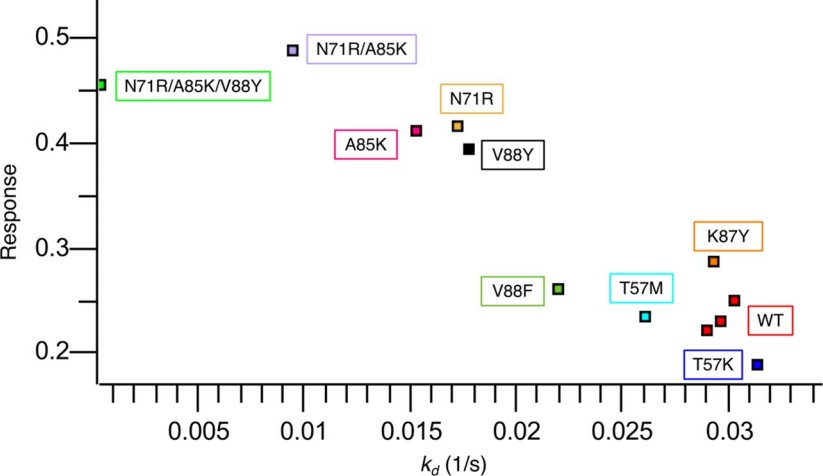

**Fig 8. Single-concentration experimental screening of c-Raf-RBD variants binding to KRas using BLI.** c-Raf-RBD variants at 250 nM were allowed to associate with KRas$^{GppNHp}$ immobilized on a Ni-NTA OctetRed96 BLI tip for 180 s and then dissociation was measured and fitted for 120 s. All dissociation fits were performed in a local 1:1 model and showed strong agreement with the data, every fit having greater than a $R^2$ of 0.99 and a $\chi^2$ lower than 0.01. The fitted dissociation rate constant ($k_d$ (1/s)) is plotted versus the response rate for each variant. Each point is labeled with its corresponding variant boxed in the corresponding color. A triplicate repeat was performed for the c-Raf-RBD wild-type (WT) variant (red). Variants fall into three groups: variants similar to WT (T57K in blue, T57M in cyan, WT in red, K87Y in orange, and V88F in forest green), variants better than WT (A85K in pink, N71R in sand, and V88Y in black), and variants with a response more than twice as large as WT (RK in purple and RKY in green). These results were used as a screen with the most promising variants being studied further by full titration BLI experiments (see Fig 10). The corresponding BLI response curves for this experiment are presented in S1 Figure.

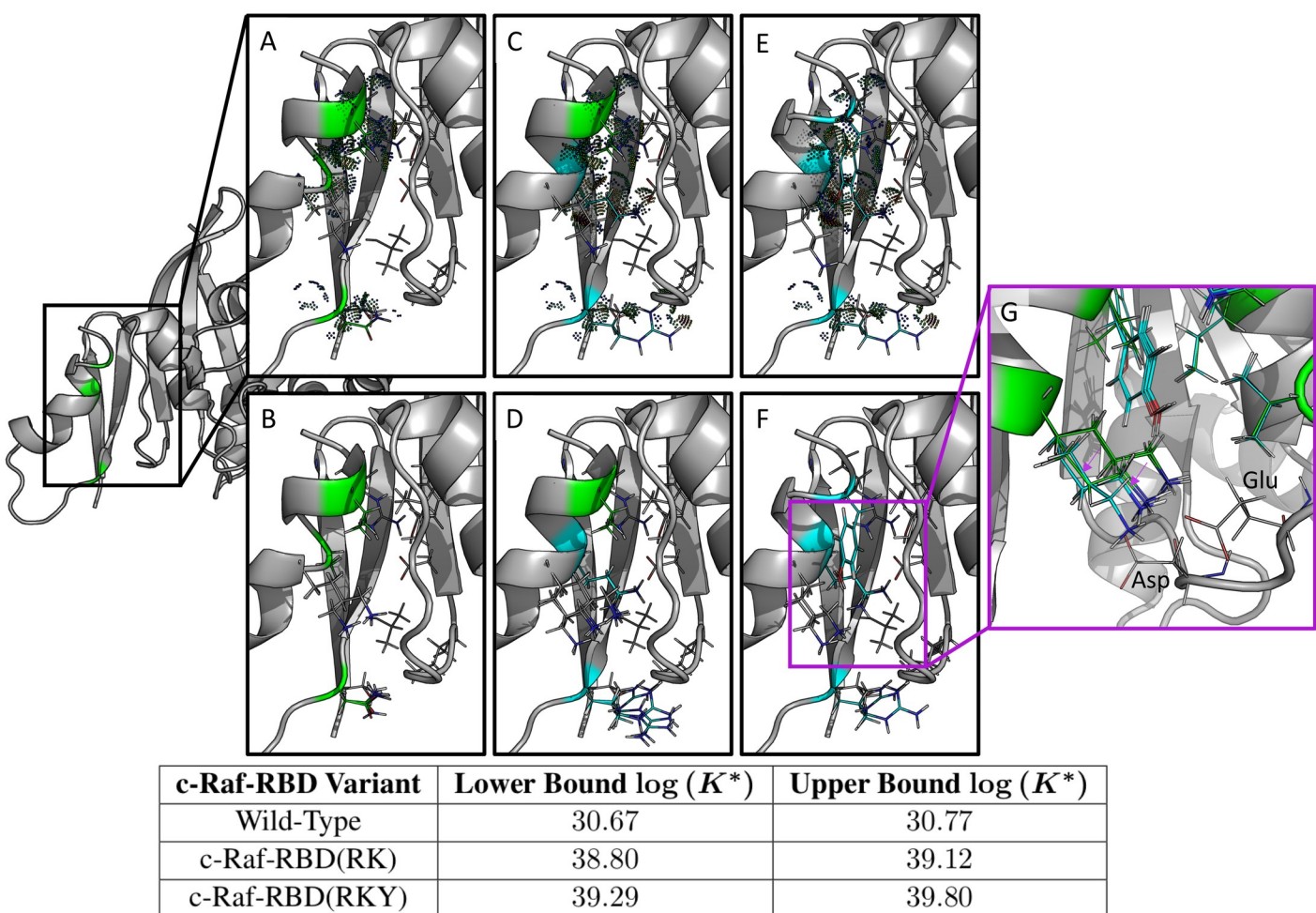

| c-Raf-RBD Variant | Lower Bound $\log(K^*)$ | Upper Bound $\log(K^*)$ |
|---|---|---|
| Wild-Type | 30.67 | 30.77 |
| c-Raf-RBD(RK) | 38.80 | 39.12 |
| c-Raf-RBD(RKY) | 39.29 | 39.80 |

**Fig 9. Computational predictions in the protein-protein interface of the c-Raf-RBD:KRas complex for c-Raf-RBD(RK) and the novel variant c-Raf-RBD(RKY).** Shown on the left is only the relevant protein-protein interface between c-Raf-RBD and KRas. Each panel zooms in on this interface and details a different c-Raf-RBD variant and its corresponding computational predictions. The upper and lower bounds on the $\log(K^*)$ score for each design variant (wild-type, c-Raf-RBD(RK), and c-Raf-RBD(RKY)) are given in the bottom table. These computational predictions correspond with and are supported by the experimental results presented in the Section entitled "Experimental validation of mutations in the c-Raf-RBD:KRas protein-protein interface." Panels (A) and (B) show the wild-type sequence, panels (C) and (D) show the variant c-Raf-RBD(RK), and panels (E) and (F) show the novel computationally predicted variant c-Raf-RBD(RKY). Panels (A), (C), and (E) show the wild-type, c-Raf-RBD(RK), and c-Raf-RBD(RKY), respectively, along with probe dots [71] that represent the molecular interactions within each structure calculated by OSPREY. These probe dots were selected to only show interactions between the residues included in the computational designs (shown as green and blue lines) with their surrounding residues. Panels (B), (D), and (F) show 10 low-energy structures from each conformational ensemble calculated by OSPREY/*EWAK\**. Panel (G) shows a zoomed-in overlay of the wild-type variant with the c-Raf-RBD variant that includes only the V88Y mutation. Purple arrows indicate the change in positioning of the lysine at residue position 84 upon mutation of residue position 88 from valine to tyrosine. When valine is present at position 88, the lysine residue (shown in green) primarily hydrogen bonds with an aspartate (labeled) in KRas. When valine is mutated to tyrosine (shown in cyan), the lysine at position 84 moves to make room for the tyrosine and positions itself to hydrogen bond with both the aspartate and the glutamate (labeled) in KRas.

A85K, and V88Y and is referred to herein as c-Raf-RBD(RKY)—in our initial BLI screen. The c-Raf-RBD(RKY) variant was computationally predicted by FRIES/*EWAK\** to bind to KRas$^{GTP}$ more tightly than the previous best known binder, c-Raf-RBD(RK) (results are detailed in Fig 9). Given the promising screening and computational predictions for the c-Raf-RBD(Y) and c-Raf-RBD(RKY) variants, we measured $K_d$ values for each variant by titrating the analyte over the ligand in a full titration BLI-based assay (see Fig 10 and the Section entitled "Bio-layer interferometry (BLI) dissociation rate and response screening" below). Titration experiments showed strong qualitative agreement with our single concentration screen. Excitingly, c-Raf-

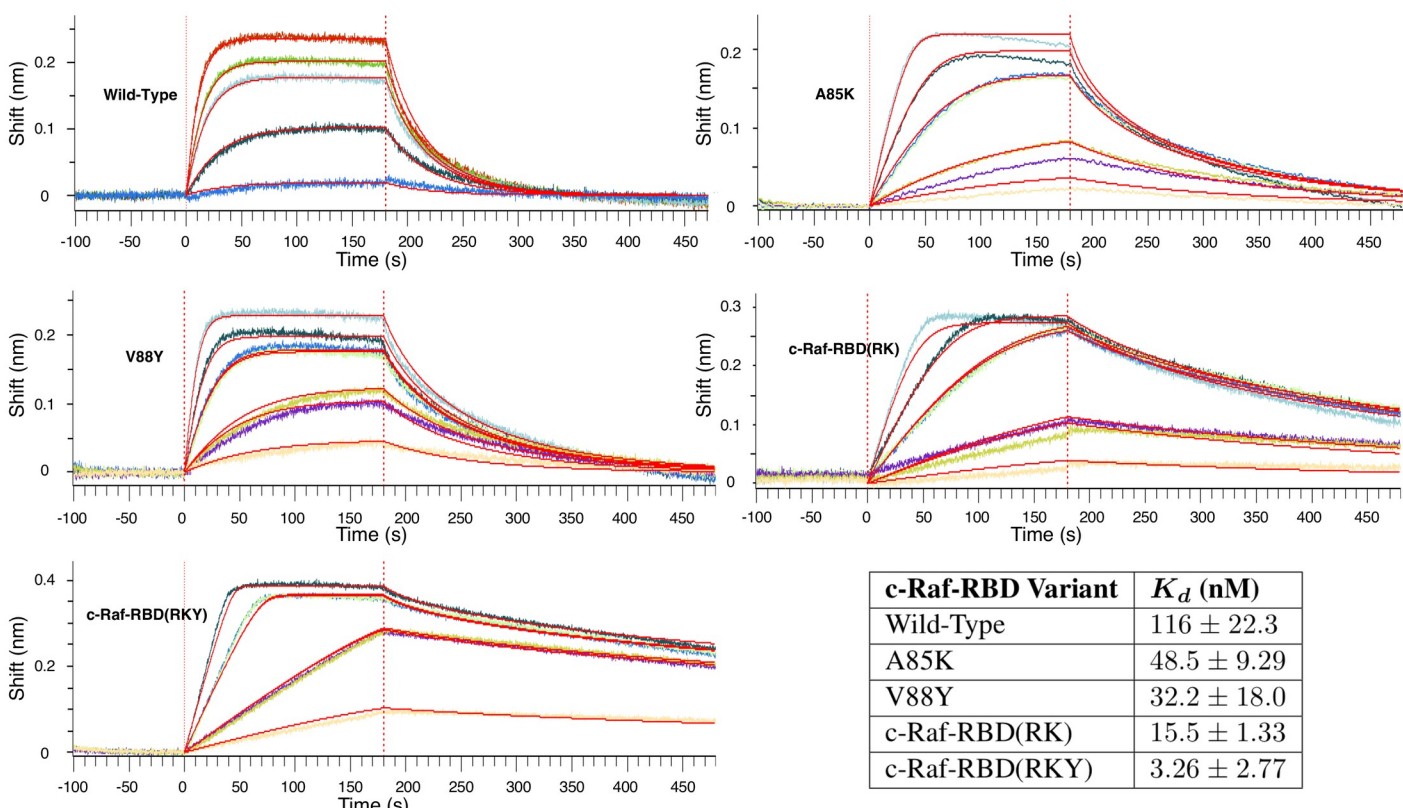

| c-Raf-RBD Variant | $K_d$ (nM) |
|---|---|
| Wild-Type | $116 \pm 22.3$ |
| A85K | $48.5 \pm 9.29$ |
| V88Y | $32.2 \pm 18.0$ |
| c-Raf-RBD(RK) | $15.5 \pm 1.33$ |
| c-Raf-RBD(RKY) | $3.26 \pm 2.77$ |

**Fig 10. BLI titration experiments to calculate $K_d$ values for select c-Raf-RBD variants. BLI titration experiments to calculate $K_d$ values for select c-Raf-RBD variants.** The plots shown here are representative and the data from replicate experiments is presented in S4 Table along with curves in S2 and S3 Figures. Each plot shows the data collected from a titration BLI experiment where the concentration of the c-Raf-RBD variant is incrementally increased. The concentrations for the wild-type variant were 10, 50, 150, 200, and 300 nM. The concentrations for all of the other variants were 10, 25, 25, 75, 75, 125, and 200 nM. Repeat intermediate concentrations were used as loading controls. These curves were then fit using a mass transport model within the Octet Data Analysis HT software provided by FortéBio in order to calculate the $K_d$ value for each variant's binding to KRas. The values in the table here (bottom right) are average $K_d$ values shown with 2 standard deviations calculated from replicate experiments (see S4 Table, S2 and S3 Figures). The values presented here for wild-type, A85K, and c-Raf-RBD(RK) agree well with previously reported $K_d$ values [60]. The best binding variant, c-Raf-RBD(RKY), binds to KRas about 5 times better than the previous tightest-known binder, c-Raf-RBD(RK), and about 36 times better than the design starting point, wild-type c-Raf-RBD.

RBD(RKY) is calculated by the data from the full titration BLI assay (see Fig 10) to bind KRas$^{GTP}$ roughly 5 times better than the previous best known binder, c-Raf-RBD(RK), and approximately 36 times better than wild-type c-Raf-RBD, the design starting point. Given how heavily studied the KRas system is, with many reported mutational and structural studies [47, 47, 48, 48, 49, 49–55, 55, 56, 56–60, 69, 70], this is a surprising discovery.

## Experimental materials and methods

Each variant of c-Raf-RBD was expressed and purified (see S2 Text) with cysteine residues at positions 81 and 96 substituted for isoleucine and methionine, respectively. These mutations were previously reported to have a minimal affect on the stability of c-Raf-RBD [55] and their substitution allows for the use of the c-Raf-RBD constructs in other assays (not mentioned herein). Additionally, we do not believe these residue substitutions have a large effect since the $K_d$ values determined herein align with previously reported $K_d$ values [60] (see Fig 10). KRas was expressed and purified (see S3 Text) with a poly-histidine protein tag (His-tag) and loaded with a non-hydrolyzable GTP analog, GppNHp. KRas was also made to include a substitution at position 118 from cysteine to serine in order to increase expression and stability [75].

**Bio-layer interferometry (BLI) dissociation rate and response screening.** His-tagged
KRas$^{GppNHp}$ was immobilized in nickel-nitrilotriacetic acid (Ni-NTA) biosensors tips and
dipped into a single concentration of 250 nM for each c-Raf-RBD variant using an Octet
Red96 instrument (FortéBio). All samples were previously diluted in kinetics buffer (PBS
[pH 7.2], 0.01% [w/v] BSA, 0.002% [v/v] Tween 20) supplemented with 200 mM NaCl, 5 mM
MgCl$_2$ and 1 mM TCEP. After steady state was achieved for all samples, samples were allowed
to dissociate in kinetics buffer (PBS [pH 7.2], 0.01% [w/v] BSA, 0.002% [v/v] Tween 20) sup-
plemented with 200 mM NaCl 5mM MgCl$_2$ and 1mM TCEP. A buffer blank and binding of
c-Raf-RBD variants to Ni-NTA tips in the absence of KRas$^{GppNHp}$ were used as references for
double subtraction. Curves (see Fig 8) were aligned on the $y$-axis to the average baseline, and
an inter-step correction was aligned to the dissociation step. A dissociation only 1:1 binding
model was used to fit the dissociation rate for a window of 120 s.

**Bio-layer interferometry (BLI) titration assay.** Binding of wild-type and variants of
c-Raf-RBD were experimentally measured using a bio-layer interferometry (BLI) titration
assay. Ni-NTA tips were then used to perform the BLI experiments to determine binding of
the c-Raf-RBD variants to KRas$^{GppNHp}$ (results along with replicates are shown in Figs 8 and
10, S4 Table, S2 and S3 Figures). All experiments were carried out in 30 mM phosphate pH
7.4, 327 mM NaCl, 2.7 mM KCl, 5 mM MgCl$_2$, 1.5 mM TCEP, 0.1% BSA, and 0.02% Tween-
20 + Kathon at 25˚C with 1000 RPM shaking and a KRas loading concentration of 20 $\mu$g/ml.
Each curve presented (see Fig 10) was fit using the built-in mass transport model within the
Octet Data Analysis HT software provided by FortéBio. We only accepted fits with a sum of
square deviations $\chi^2$ less than 1 (FortéBio recommends a value less than 3) and a coefficient of
determination $R^2$ greater than 0.98.

## Discussion

FRIES and $EWAK^*$ are new, provable algorithms for more efficient ensemble-based computa-
tional protein design. Efficiency and efficacy were tested and shown across a total of 2,826 dif-
ferent design problems. An implementation of FRIES/$EWAK^*$ is available in the open-source
protein design software OSPREY [1] and all of the data has been made available (see Data
Availability Statement). FRIES/$EWAK^*$ in combination achieved a significant runtime improve-
ment over the previous state-of-the-art, $BBK^*$, with runtimes up to 2 orders of magnitude
faster. $EWAK^*$ also limits the number of minimized conformations used in each $K^*$ score
approximation by up to about 2 orders of magnitude while maintaining provable guarantees
(see the Section entitled "Energy Window Approximation to K$^*$ ($EWAK^*$)"). FRIES alone is
capable of reducing the input sequence space while provably keeping all of the most energeti-
cally favorable sequences (see the Section entitled "Fast Removal of Inadequately Energied
Sequences (FRIES)"), decreasing the size of the sequence space by more than 2 orders of magni-
tude, and leading to more efficient design given the smaller search space.

To further validate OSPREY with FRIES/$EWAK^*$, we applied these algorithms to a well-studied
and biomedically interesting system: the c-Raf-RBD:KRas PPI. First, we performed a series
of retrospective designs where FRIES/$EWAK^*$ accurately predicted how a variety of mutations
affect the binding of c-Raf-RBD to KRas$^{GTP}$ that previous computational methods had failed
to accurately predict [60]. This success supports the use of OSPREY and FRIES/$EWAK^*$ to evaluate
the affect mutations in the protein-protein interface of c-Raf-RBD:KRas have on binding
(more, similar successes of the $K^*$ algorithm are presented and discussed in [1]). FRIES/$EWAK^*$
also prospectively predicted the effect of new mutations in the c-Raf-RBD:KRas PPI and dis-
covered a novel c-Raf-RBD mutation V88Y with improved affinity for KRas. We went on to
combine this new mutation with two previously reported mutations, N71R and A85K [60], to

create c-Raf-RBD(RKY), an even stronger binding c-Raf-RBD variant, which FRIES/*EWAK*\* accurately predicted. We biochemically screened top predicted variants using an initial bio-layer interferometry (BLI) single-concentration assay. Only a promising subset of the computationally predicted and initially screened variants were then evaluated using a BLI titration assay to calculate $K_d$ values for individual c-Raf-RBD variants. We determined that c-Raf-RBD (RKY) binds to KRas$^{GTP}$ roughly 36 times more tightly than wild-type c-Raf-RBD, making it the tightest known c-Raf-RBD variant binding partner of KRas$^{GTP}$.

Given that numerous groups have explored this protein-protein interaction [47–59] and performed mutagenesis on c-Raf-RBD either, through rational means [47, 48, 56, 69], computational methods [49, 60] or high-throughput evolutionary methods [55, 70] and that none identified V88Y, this discovery validates our computational approach and the use of computational algorithms such as FRIES and *EWAK*\* to re-design protein-protein interfaces toward improved binding. Additionally, previous mutations that enhanced the affinity of c-Raf-RBD to KRas did so by supercharging c-Raf-RBD [48, 49, 60]. In contrast, our mutation V88Y introduces a novel, aromatic residue. The discovery that such a mutation can improve the binding of c-Raf-RBD to KRas$^{GTP}$ suggests that previous work has not completely explored the sequence space available to this binding interaction. These new c-Raf-RBD variants could be fused to cell-penetrating peptides and used as in-cell tools to further characterize KRas:effector signalling.

## Supporting information

**S1 Text. Homology model of c-Raf-RBD in complex with KRas.**
(PDF)

**S2 Text. Details of the expression and purification of c-Raf-RBD variants.**
(PDF)

**S3 Text. Details of the expression and purification of KRas.**
(PDF)

**S4 Text. Sequence and partition function approximation algorithms.**
(PDF)

**S5 Text. The relationship between the energy window stopping criterion and epsilon.**
(PDF)

**S1 Table. Protein structures used in computational experiments as described in the Section entitled "Computational materials and methods".** Each protein structure has its PDB ID listed along with its molecule names as presented in the Protein Database entry for each structure. Individual designs are not listed or described here, but the necessary code and data is provided for the interested reader (see Data availability statement).
(PDF)

**S2 Table. Experimental and computational percent change in binding and rankings.** For each listed c-Raf-RBD variant, we give the experimental percent change in KRas binding relative to wild-type c-Raf-RBD as reported in [48] (no $K_a$ values were reported in [48] so the corresponding entries are left blank here) and as calculated from reported binding values in [54] and [60] (reported here as Exp. $K_d$), the *EWAK*\* computationally predicted percent change in binding, the Δb values as described in the Section entitled "FRIES/EWAK\* retrospectively predicted the affect mutations in c-Raf-RBD have on binding to KRas," and the rankings that correspond to these values. The Δb values are calculated as follows: $\log_{10}(\%) - 2$ where %

represents the percent change in binding upon mutation. The rankings have a Pearson correlation of 0.81. The Pearson correlation between the change in binding values Δb is 0.64.
(PDF)

**S3 Table. Table of computational predictions for point mutants in c-Raf-RBD.** Each section of the table shows the results of the redesign of a residue position in c-Raf-RBD in the c-Raf-RBD:KRas PPI in order of increasing upper bound on $\log(K^*)$. The table contains the values for upper and lower bounds on $\log(K^*)$ values (these bounds are described in detail in [32]). *Design results for the wild-type amino acid identity for each position. †Mutations that were selected for experimental testing and validation.
(PDF)

**S4 Table. $K_d$ values for each tested variant for all replicates of BLI titration experiments.** For each listed variant, we give the dissociation constant $K_d$ for each BLI titration experiment calculated from the fit done using the built-in mass transport model within the Octet Data Analysis HT software provided by FortéBio. We only accepted fits with a sum of square deviations $\chi^2$ less than 1 (FortéBio recommends a value less than 3) and a coefficient of determination $R^2$ greater than 0.98. Presented in the table in Fig 10 are averages of these $K_d$ values.
(PDF)

**S1 Figure. Curves for single concentration BLI screen of c-Raf-RBD variants.** c-Raf-RBD variants at 250 nM were allowed to associate with KRas$^{\text{GppNHp}}$ immobilized on a Ni-NTA OctetRed96 BLI tip for 180 s and then dissociation was measured and fitted for 120 s. All dissociation fits were performed in a local 1:1 model and showed strong agreement with the data, every fit having greater than a $R^2$ of 0.99 and a $\chi^2$ lower than 0.01. Each curve is labeled with its corresponding c-Raf-RBD variant boxed in the matching color. A triplicate repeat was performed for the c-Raf-RBD wild-type (WT) variant (Red). Curves grouped into three groups: variants similar to WT (T57K in blue, T57M in cyan, WT in red, K87Y in orange, and V88F in forest green), variants better than WT (A85K in pink, N71R in sand, and V88Y in black), and variants with a response greater than twice that of the WT (RK in purple and RKY in green).
(PDF)

**S2 Figure. Replicate BLI titration curves of c-Raf-RBD(RKY) binding to immobilized KRas on NiNTA tips.** Titration experiments were conducted over different concentration ranges and for different association and dissociation times in order to avoid artifacts. Within each titration experiment, curves were fit globally to a mass transport model using the FortéBio Data Analysis HT software. All fits achieved an $R^2$ greater than 0.99 and a $\chi^2$ smaller than 0.65. The two titration experiments on the left are replicates with concentrations ranging from 150 nM to 4.69 nM in a 2-fold serial dilution. The titration experiment on the top right has titrations ranging from 150 nM to 9.38 nM in a 2-fold serial dilution but with an extended association step. The titration in the bottom right contains binding curves with the following concentrations of c-Raf-RBD(RKY): 200 nM, 125 nM, 75 nM, 75 nM, 25 nM, 25 nM, and 10 nM. Note the in-experiment repetition of two concentrations (75 nM and 25 nM). This was done in order to control for response and curve shape within an experiment. Curves for the repeat concentrations show strong reproducibility and alternating what repeat curves are used for the global fit changes the $K_d$ within a range of 1.99 nM to 2.34 nM. Results from these four titration experiments were averaged to generate a dissociation constant and standard deviation for c-Raf-RBD(RKY). Results are reported in the manuscript as the dissociation constant ± two standard deviations.
(PDF)

**S3 Figure. Replicate BLI titration curves of c-Raf-RBD(RK) binding to immobilized KRas on NiNTA tips.** Titration experiments were conducted over different concentration ranges and for different association and dissociation times in order to avoid artifacts. Within each titration experiment, curves were fit globally to a mass transport model using the FortéBio Data Analysis HT software. All fits achieved an $R^2$ greater than 0.98 and a $\chi^2$ smaller than 0.25. The titration experiment on the top left was done with the following concentrations of c-Raf-RBD(RK): 200 nM, 125 nM, 75 nM, 75 nM, 25 nM, 25 nM, and 10 nM. Note the in-experiment repetition of two concentrations (75 nM and 25 nM). This was done in order to control for response and curve shape within the experiment. Curves for the repeat concentrations show strong reproducibility and alternating what repeat curves are used for the global fit changes the $K_d$ within a range of 15.1nM to 15.48nM. The bottom left and top right titration experiments are replicates with concentrations ranging from 150 nM to 4.69 nM in a 2-fold serial dilution. Results from these three titration experiments were averaged to generate a dissociation constant and standard deviation for c-Raf-RBD(RK). Results are reported in the manuscript as the dissociation constant ± two standard deviations.
(PDF)

## Acknowledgments

We thank Rachel Kimbrough, Chanelle Simmons, Catherine Ehrhart, and Kelly Huynh for assistance with experiments and biochemical validation, Ben Fenton and Michael Kennedy for the initial KRas plasmids, and Paul Modrich for the Rosetta 2(DE3) cell lines. We also thank Terrence Oas and all members of the lab for helpful discussions.

## Author Contributions

**Conceptualization:** Anna U. Lowegard, Marcel S. Frenkel, Graham T. Holt, Jonathan D. Jou, Adegoke A. Ojewole, Bruce R. Donald.

**Data curation:** Anna U. Lowegard, Marcel S. Frenkel, Bruce R. Donald.

**Formal analysis:** Anna U. Lowegard, Marcel S. Frenkel, Bruce R. Donald.

**Funding acquisition:** Anna U. Lowegard, Bruce R. Donald.

**Investigation:** Anna U. Lowegard, Marcel S. Frenkel, Bruce R. Donald.

**Methodology:** Anna U. Lowegard, Marcel S. Frenkel, Graham T. Holt, Jonathan D. Jou, Adegoke A. Ojewole, Bruce R. Donald.

**Project administration:** Bruce R. Donald.

**Resources:** Bruce R. Donald.

**Software:** Anna U. Lowegard, Marcel S. Frenkel, Graham T. Holt, Jonathan D. Jou, Adegoke A. Ojewole, Bruce R. Donald.

**Supervision:** Bruce R. Donald.

**Validation:** Anna U. Lowegard, Marcel S. Frenkel, Bruce R. Donald.

**Visualization:** Anna U. Lowegard, Marcel S. Frenkel, Bruce R. Donald.

**Writing – original draft:** Anna U. Lowegard, Bruce R. Donald.

**Writing – review & editing:** Anna U. Lowegard, Marcel S. Frenkel, Graham T. Holt, Jonathan D. Jou, Bruce R. Donald.

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
