## [Decision Letter · Decision Letter 0]

6 Dec 2019

Dear Dr Donald,

Thank you very much for submitting your manuscript 'Novel, provable algorithms for efficient ensemble-based computational protein design and their application to the redesign of the c-Raf-RBD:KRas protein-protein interface' for review by PLOS Computational Biology. Your manuscript has been fully evaluated by the PLOS Computational Biology editorial team and in this case also by independent peer reviewers. The reviewers appreciated the attention to an important problem, but raised some substantial concerns about the manuscript as it currently stands. While your manuscript cannot be accepted in its present form, we are willing to consider a revised version in which the issues raised by the reviewers have been adequately addressed. We cannot, of course, promise publication at that time.

As you can see, the two reviewers have quite contrasting views of the paper. If you decide to submit a revision, it is likely that we will involve a third reviewer. The first reviewer has valid criticisms of the experimental data and suggests additional computational benchmarking designed to test what the method is actually trying to achieve.

Sincerely,

Roland L. Dunbrack Jr., Ph.D.

Associate Editor

PLOS Computational Biology

Nir Ben-Tal

Deputy Editor

PLOS Computational Biology

[LINK]

Reviewer's Responses to Questions

**Comments to the Authors:**

Reviewer #1: Lowegard et al. present a development of the Donald lab Osprey design methodology to improve efficiency in the design of large sequence ensembles. If I understood correctly, Osprey carries out a combined sequence-design and backbone relaxation step and therefore requires enumerating sequence space. To reduce the enumerated space to a size that can be practically computed, they now compute the stability of each component of the protein system (for instance, receptor and ligand) to ensure that no mutant is too destabilising before considering all combined mutations. The authors tested their approach on a natural (and engineered) PPI involving KRas and retrospectively tested the ranking of mutants relative to previous mutational analyses. They also used the method to design a single-point mutant and found that it improved affinity fivefold relative to the starting point.

Major concerns:

1. The main message of the paper is rather difficult to distill. I wrote above how I understood the motivation for the current paper and how the authors addressed the computational problem, but this not stated in the abstract and it was only by going through the methods that I could understand this (and I'm still not sure that I understood correctly). The methods are, as expected for such a paper, quite dense with formulae so the big picture is lost on a reader. My first suggestion is to clarify already in the abstract what is the main contribution of this paper, not just by stating that the method is more efficient, but why it is more efficient and for which problems. In this connection, the provability of the method is of far less significance to most users and developers of computational design methodology than its practical usefulness (accuracy, speed, scope). Nevertheless, the point about provability is repeated from the second word of the title to the end of the paper in the excess of 30 times. My second suggestion is therefore to substantially reduce the emphasis from this point and highlight applicability and accuracy.

2. The authors chose the KRas system apparently because it is a drug target. They mention several times that KRas is undruggable and that this is therefore a biomedically significant problem. First, as the authors mention at one point in the text, KRas is actually not undruggable and there are now small molecules in clinical trials (or in the clinic; I'm not sure). Second, whether or not KRas is a drug target is beside the point for this paper, since KRas is an intracellular target and no matter what affinities the authors achieved, the designs could not be used in any conceivable way in treatments or even in drug discovery. I therefore recommend that they drop all reference to druggability and concentrate on the truly important aspect, which is that KRas has been studied extensively as a model PPI, thus providing an excellent data set for retrospective analysis.

3. The narrative of this study is confusing: the authors developed new methods to enable design within large sequence spaces, but in the end, they validated their method by testing it retrospectively against a predetermined set of mutants and in the prospective design of a single-point mutation. It seems that any molecular threading approach and the simplest mutational scanning method (FOLDX?) would be just as useful for this analysis as the newly developed method. This is, in my opinion, a critical point. I don't think that the validation presented here is sufficient and instead, the authors should show that in the design of large ensembles, they recapitulate known sequence signatures (for instance, natural sequence alignments), and they should do this for more than one protein. Such a benchmark would show that the method is indeed fast enough to be practically useful for large sequence spaces and that it yields more than anecdotal successes.

4. Regarding the single-point mutant that the method prospectively designed. This mutant showed at most fivefold improvement in affinity over the starting point engineered variant. This is quite modest to say the least, though the authors trumpet this result as "a discovery of some significance". The tone relating to this mutant should be reduced throughout the paper.

5. The BLI experimental methods are described far too briefly to understand what was actually done and what fits are reported. It is not clear what model was used to fit the data. It looks to me like the authors may have fitted a kinetic model but I'm not sure whether they fitted each curve to a separate kinetic model or all curves at once (the latter is the correct way of analysing these data). The fits to the RK variant (Fig10) are quite poor. Considering that they have 6 independent measurements, and 4 of them show poor fits, I think that this experiment needs to be redone and since this serves as the baseline measurement to judge the impact of the VY designed mutation, this is quite important. I also suggest to draw the fits in black because some of the traces are quite close in tone to the red used to show the fits.

6. Also, typically when such modest improvements in affinity are reported (fivefold), and given that the fits are rather poor in the data that were presented, it's important to provide experimental replicates. It is difficult to say whether the fivefold effect is real or within the noise of the experimental setup.

Minor comments:

1. Eq. 2: define C,P,L

2. On pg. 17, the authors use Spearman correlations to check the rank order correlation in 38 mutations. Since the computational method reports energies and the binding experiments report KDs, why not use Pearson correlations?

3. Line 403: the authors state that they filtered mutations based on "promising K* scores and structure examination". It's important that they provide some guidelines about how they selected the mutation. This explanation cannot be reproduced by anyone.

4. The single-concentration BLI measurements are very unusual (Fig8). I'm not sure that they are meaningful at all since at a single concentration it's impossible to determine Kd and it's not very obvious what is the signal one measures. I recommend to drop this analysis as it is misleading, especially since the authors draw conclusions on the correlation between these experimental results and the computational analysis.

5. Lines 433-437: The authors make a lot of the fivefold improvement that the single point mutation exhibited. It's a quite modest improvement and seems almost anecdotal given that no other mutants were presented.

6. Lines 439-452: this belongs in methods.

7. 475: "we applied these algorithms to a biomedically significant design problem": this is not a biomedically significant design problem, because the design cannot be used in a biomedical context. KRas is biomedically significant drug target but the design is in no way a starting point for a drug. Again, the authors can highlight the datasets that are available for KRas but not the druggability aspects as they have no relevance to the work.

8. 502: "the discovery that such a mutation can improve the binding...is of considerable significance...eventually developing successful therapeutics". This is quite spurious. They tested a single mutation which exhibited a very modest effect and has no relevance to drug discovery.

Reviewer #2: The authors present two algorithmic improvements for the K* algorithm for predicting binding affinities at protein/protein interfaces, and then show that their improved algorithm is capable of accurately predicting binding energies with a retrospective analysis of mutations at the cRaf-RBD / KRas interface and by predicting several point mutations that improve binding relative to wild-type, including one mutant, V88Y, that when paired with two previously-reported mutations, creates the tightest-yet-known interface between these two proteins. The paper spans from theory to the bench and is an impressive piece of work.

The two algorithmic improvements the authors describe focus on the two levels at which the K* algorithm broaches an exponential amount of work, 1) at the conformation level, and 2) at the sequence level.

At the conformation level, the K* algorithm is designed to approximate K = q_pl / ( q_p x q_l ) by enumerating all conformations of a single sequence in order of increasing energy until the conformations remaining are at a high-enough energy that their contribution to the partition function is small -- small enough that the resulting K* approximation is within a user-provided constant, epsilon, of the full K. To improve performance here, the authors introduce the EWAK* algorithm. EWAK* instead of enumerating conformations until the epsilon-error threshold is reached, they stop when the bound on the energy of the next conformation is larger than some user-provided deltaE of the best conformation. The authors contrast EWAK* with the previous algorithm, BBK*, and note that it produces significant performance improvements.

I found this section a little difficult to understand: A) The algorithmic improvement presented in the BBK* paper from 2017 focuses entirely on sequence-space pruning and not conformation-space pruning, and so EWAK* seems like it ought to be compared to K* (or that the text should say that BBK* and K* are the same for the sake of this comparison). B) It would be nice to understand how the user providing an energy threshold "1 kcal/mol" differs from the user changing epsilon to ".05" from "0.01". Is there a simple mathematical relationship between these numbers or are they related but not comparable?

2) At the sequence level, the authors present an algorithm, FRIES, for removing sequences that are higher in energy than the wild type sequence. Here both FRIES and BBK* enumerate sequences in order of decreasing bounds on their energies. With FRIES, the intuition on when to stop is that, after the wild-type sequence is encountered at an interface where one is looking for tighter binding, then there is not much point in continuing.

I found this section slightly confusing because it's not clear whether the energies are for the complex structure, the unbound structure, or both. It's also unclear why FRIES searches the multi-sequence tree until the wild type sequence is hit instead of going directly to the (clearly known!) wild type sequence. I believe what the authors mean to describe is that FRIES searches the multi-sequence tree and descends into the single-sequence conformation tree for each sequence it encounters _until_ it hits the wild type sequence, after which point it begins looking to stop sequence enumeration.

I would also be curious to understand why the authors chose to approximate q^(-)_wt using only a single conformation of the wild-type sequence instead of trying to estimate q*_wt accurately; the wild type sequence is just one more sequence among the very many sequences that FRIES and BBK* would enumerate.

Minor point: on page 5 the phrase "by up to more than 2 orders of magnitude" confuses me; is it "by up to 2 orders of magnitude" or "more than 2 orders of magnitude"?

**Have all data underlying the figures and results presented in the manuscript been provided?**

Reviewer #1: None

Reviewer #2: Yes

PLOS authors have the option to publish the peer review history of their article (what does this mean?). If published, this will include your full peer review and any attached files.

Reviewer #1: No

Reviewer #2: No

---

## [Decision Letter · Decision Letter 1]

11 Apr 2020

Dear Dr. Donald,

Thank you very much for submitting your manuscript "Novel, provable algorithms for efficient ensemble-based computational protein design and their application to the redesign of the c-Raf-RBD:KRas protein-protein interface" for consideration at PLOS Computational Biology. As with all papers reviewed by the journal, your manuscript was reviewed by members of the editorial board and by several independent reviewers. The reviewers appreciated the attention to an important topic. Based on the reviews, we are likely to accept this manuscript for publication, providing that you modify the manuscript according to the review recommendations.

Both reviewers recommended publication. The second reviewer had some concerns about whether all the necessary data are presented in the paper. S/he suggests some work that might be outside the scope of the paper, and we think that does not need to be included in your final revision. But please respond to the other suggestions in a minor revision.

Sincerely,

Roland L. Dunbrack Jr., Ph.D.

Associate Editor

PLOS Computational Biology

Nir Ben-Tal

Deputy Editor

PLOS Computational Biology

[LINK]

Reviewer's Responses to Questions

**Comments to the Authors:**

Reviewer #1: Some of the criticisms I raised before are still relevant to this revision: prospective validation is done on just one single-point mutant and the results are modest compared to data presented in recent years for protein design methods. Nevertheless, the authors made a very serious effort to clarify the message and provide more detail on the calculations and the experiments while also toning down some of the language that seemed be carried away in the original submission.

I think that the major contribution of this paper is in the very extensive theoretical treatment. Further experimental validation including, possibly, a side-by-side comparison of this method with others may provide the answer to the questions I raised on the method's scope.

In summary, I would like to congratulate the authors on this work and also on their sincere efforts to address my previous comments which may have been phrased too harshly (my apologies for that).

Reviewer #2: I very much like this paper and would like to see it published soon. I think there are a handful of relatively small things that should be addressed that do not necessarily require another round of review.

I think the objections made by the first reviewer to the original submission were well deserved and that the changes that were made to the text in response have improved the manuscript considerably. Removing the repetitive emphasis on provability made the paper much more enjoyable to read.

I also think the point about using Spearman's rho instead of Pearson's r is still important and was not well addressed in the revision: the authors are not reporting r presumably because the T68K outlier makes r look quite bad. I can only guess that the authors suspect a reader would distill the paper to a single r value and look past Osprey as a tool for their project. I wanted to compute an r value myself looking to the data in Supp Table 3, but while the upper and lower bounds on log(K) for each mutation + wt is given, that does not readily translate into the delta-b value that is presented in Figure 5. Another column for delta-b would be nice. Also missing from Supp Table 3 is the WT dG values (where available) that a reader would need to compute r accurately.

A paper where the authors look into how well K* with Osprey's energy function performs compared to other techniques for binding energy calculations (e.g. FoldX) is definitely needed, but perhaps is beyond the scope of this work. On this topic, I would also add that I find the sentence explaining why K* does not do a good job predicting absolute values of binding energies

"since our current designs likely underestimate entropic contributions to binding upon mutation due to various limitations in biological modeling"

to be very unsatisfying! I was under the impression that K* would do a much better job than other techniques for estimating mutation ddGs because it considers side-chain entropic contributions explicitly. If entropy is poorly estimated by EWAK*, wouldn't it be estimated even more poorly by other molecular modeling applications? Finally, what aspect of entropy do you suspect is under-considered? ("various" doesn't help me). One question that perhaps could be addressed in a subsequent ddG comparison paper is: how well does K* (including FRIES and EWAK) perform using a different energy function? (Can the energy function that Osprey is using be swapped out with another energy function such as the one used in FoldX?)

I appreciate the clarity that was added to the algorithm description, especially with regard to fact that the partition functions of the three species are approximated separately, and with the energy bound based on the WT sequence that FRIES uses to prune sequences.

The revision was missing almost all of the figures -- only figure 8 of the non-supplemental-figures is included. I had to go back to the original manuscript to find the others.

**Have all data underlying the figures and results presented in the manuscript been provided?**

Reviewer #1: None

Reviewer #2: No: The data needed to reconstruct figure 5 is not fully present in supplemental table 3. It is possible for the reader to hunt down all of the previously reported experimentally measured ddGs from the previous literature, but it would be nice for the authors to simply include these numbers. It is less clear how one goes from the upper and lower bounds on the log(K) to the delta-b values that the authors use.

PLOS authors have the option to publish the peer review history of their article (what does this mean?). If published, this will include your full peer review and any attached files.

Reviewer #1: No

Reviewer #2: No
---

## [Editor Report · Decision Letter 2]

13 May 2020

Dear Dr. Donald,

We are pleased to inform you that your manuscript 'Novel, provable algorithms for efficient ensemble-based computational protein design and their application to the redesign of the c-Raf-RBD:KRas protein-protein interface' has been provisionally accepted for publication in PLOS Computational Biology.

Thank you for taking into account the comments made by the reviewers on the revised manuscript, and especially for providing the additional data requested.

Best regards,

Roland L. Dunbrack Jr., Ph.D.

Associate Editor

PLOS Computational Biology

Nir Ben-Tal

Deputy Editor

PLOS Computational Biology

---

## [Editor Report · Acceptance letter]

2 Jun 2020

PCOMPBIOL-D-19-01654R2 

Novel, provable algorithms for efficient ensemble-based computational protein design and their application to the redesign of the c-Raf-RBD:KRas protein-protein interface

Dear Dr Donald,

I am pleased to inform you that your manuscript has been formally accepted for publication in PLOS Computational Biology. Your manuscript is now with our production department and you will be notified of the publication date in due course.

With kind regards,

Laura Mallard
